# Metabolome-driven microbiome assembly in ginger (*Zingiber officinale*) enhances nutrient cycling and crop yield through keystone taxa

Wenbo Wang[1], Wenxing He[1], Yaoxing Zhang ●[1], Xia Wang[1], Jialin Li[1], Xiujun Zhang[1], Beibei Chu[2], Yanshun Nie[3], Nayanci Portal-Gonzalez ●[1] ✉ & Ramon Santos-Bermudez ●[1] ✉

Plant microbiomes shape crop performance, but the mechanisms by which host-derived metabolites influence the recruitment and organisation of beneficial microbes—and how these affect crop yield—remain poorly understood. Few studies have linked multi-kingdom microbiome structure, metabolite profiles, and agronomic traits under field conditions. We integrated 16S rRNA and ITS amplicon sequencing with untargeted LC-MS/MS metabolomics across 36 samples from two *Zingiber officinale* (ginger) cultivars with contrasting yields. Bacterial communities were primarily shaped by stochastic processes (neutral model $R^2 = 0.67–0.68$), while fungal assembly was deterministic (βNTI < −2 in roots and rhizomes). The high-yield cultivar exhibited more complex co-occurrence networks (596 vs. 272 edges) and enrichment of keystone taxa, including *Talaromyces* and *Devosia*. Metabolomic profiling identified 586 unique compounds, with 24 enriched pathways in the high-yield cultivar, notably isoflavonoid biosynthesis and folate metabolism. Key host metabolites—Niazimin A and 1-oleoyl-lysophosphatidic acid—showed strong positive correlations ($r > 0.75$, $P < 0.01$) with nitrogen-fixing and growth-promoting microbes, whereas Oxindole correlated negatively. These findings suggest that host metabolic shifts and keystone microbes co-regulate microbiome structure and nutrient cycling. Our results provide mechanistic insight into microbiome-mediated yield differences and a basis for microbiome-informed crop design.

The plant microbiome is essential for several vital functions: nutrient acquisition, immune regulation, pathogen suppression, and stress mitigation[1–3]. As a key factor in determining crop resilience and productivity, these microbial communities—comprising bacteria, fungi, and archaea—create complex networks within various plant-associated environments such as soil, the rhizosphere, and internal tissues[4,5]. Microbiome composition and function are shaped by multiple interacting forces: host genotype, environmental factors, microbial interactions, and chemical signalling through metabolites[6]. While advances in metagenomics and metabolomics have revealed complex plant–microbe interactions in major crops, our mechanistic understanding of microbiome assembly, especially in high-value, non-model plants, remains limited[7].

Ginger (*Zingiber officinale* Roscoe), a perennial crop of global importance for culinary and medicinal applications, suffers from productivity losses primarily due to soilborne diseases such as rhizome rot[8,9]. In two recent studies[10,11], we provided initial insights into the microbiome and metabolome of ginger under health and disease conditions. In Wang et al.[10], we profiled the rhizosphere and endophytic microbiota of healthy vs. diseased ginger plants. We showed that specific microbial taxa—including *Pectobacterium* and *Plectosphaerella*—were enriched in diseased samples, while beneficial genera like *Bacillus* and *Mortierella* were associated with healthy plants. Additionally, we identified several plant metabolites, such as 1-oleoyl lysophosphatidic acid and Niazimin A, that correlated with health-associated microbiota. This study suggested that specific metabolites may

[1]School of Biological Science and Technology, University of Jinan, Jinan, Shandong, People's Republic of China. [2]Agriculture and Rural Affairs, Bureau of Changyi, Weifang, Shandong, People's Republic of China. [3]Fengtang Ecological Agriculture Technology Research and Development (Shandong) Co.Ltd., Taian, China. ✉e-mail: bio_nayanci@ujn.edu.cn; bio_ramon@ujn.edu.cn

act as microbial selectors, influencing the recruitment of protective or pathogenic taxa.

In a complementary study[11], we compared metabolite and microbial profiles of ginger cultivated under open-field versus greenhouse conditions. We identified over 500 metabolites, with key differences in pathway enrichment (e.g., isoflavonoids, amino acid derivatives) based on environmental context. This study highlighted how the agricultural environment shapes both the composition of microbial communities and the buildup of metabolites. This finding points to a three-way interaction involving plant genotype, environmental conditions, and the microbiome. However, neither study directly addressed how these interactions impact crop yield, nor did they resolve the ecological processes (e.g., deterministic vs. stochastic assembly) governing microbiome composition across different plant compartments.

What remains unclear is (1) how microbial assembly mechanisms differ between bacterial and fungal communities in ginger; (2) whether host-derived metabolites act as deterministic filters shaping high-yield-associated microbiomes; and (3) which keystone taxa mediate the observed differences in nutrient cycling and microbial network complexity across cultivars with distinct productivity.

To fill these gaps, the present study integrates 16S rRNA and ITS amplicon sequencing with untargeted LC–MS/MS metabolomics across 36 samples from two ginger varieties differing in yield. Unlike our previous studies[10,11], which focused on disease-associated microbiota and environmental metabolite variation, this work explicitly investigates the ecological drivers of microbial community assembly (e.g., deterministic vs. stochastic processes), identifies metabolite–microbe correlations linked to productivity, and defines keystone taxa that underlie microbial network complexity and nutrient cycling. By uncovering how host-derived metabolites act as selective filters and how network topology differs between cultivars, we provide a mechanistic framework for microbiome-mediated yield enhancement. This foundational knowledge paves the way for microbiome-based strategies to improve sustainability[12] in ginger cultivation and other high-value crops.

## Methods
### Experimental design and sampling
Samples of *Zingiber officinale* var. officinale (variety one) were collected from Laiwu District in Jinan, Shandong Province, China (36°19′50″ N, 117°29′29″ E). This region has a warm-temperate, humid, semi-humid climate with sandy loam soil[9]. In contrast, samples of *Zingiber officinale* var. amarum (variety two) were collected from Changyi District in Weifang, Shandong Province (36°52′ N, 119°24′ E), which features a temperate monsoon semi-humid climate with cinnamon soil[13]. Each sampling plot covered approximately 666 square metres and contained 7000 and 8000 ginger plants. No other crops were grown within a 1500-metre radius of the sampling sites to minimise cross-contamination. The cultivation process involved ten irrigation events and the application of 100 kilograms of both compound and organic fertilisers at critical growth stages. Both varieties were managed following the same agronomic protocol to reduce location-related variability, which included standardised fertiliser and irrigation regimes, equal planting density, and synchronised sampling at comparable developmental stages. The plots were cultivated without intercropping, and all samples were collected within the same week to avoid seasonal biases.

Sampling was conducted from December 15 to 18, 2021, utilising three biological replicates randomly selected from adjacent plots for each variety under examination. Each biological replicate consisted of a composite sample that included homogenised individual plant compartments - leaf, stem, root, and rhizome - along with the associated soils, explicitly targeting the rhizosphere and bulk soil collected from three separate plants. Plants were spaced at least 3 metres apart to ensure each replicate's independence, and care was taken to select specimens exhibiting similar size and developmental stages. Thirty-six biological (composite) samples were systematically harvested for each variety. Following established protocols[10], all

samples were promptly stored at -80 °C to maintain their integrity before DNA extraction.

### Soil and plant physicochemical properties and assessment of crop yield
Soil physicochemical properties were analysed, including pH, organic matter, total potassium, nitrogen, phosphorus, and nutrient availability. Soil organic matter was measured using the potassium dichromate oxidation technique, while available nitrogen was assessed via alkaline hydrolysis diffusion[14]. Total nitrogen content was quantified using a dry combustion elemental analyser (Vario EL III, Germany) and further analysed with a carbon and nitrogen element analyser (Leco CNS-2000, Leco Corporation, St. Joseph, MI, USA). Total phosphorus and potassium levels were determined through combustion (2400 I, CHNS/O PerkinElmer, Boston, MA, USA), with phosphorus quantified via molybdenum-antimony colourimetry. Potassium content was measured using a flame photometer, and available phosphorus was extracted using the NaHCO3 method[15]. Plant protein content was analysed using the Kjeldahl method. Soil pH was determined by mixing samples with deionised water at a 1:2.5 ratio (m/v)[16] (see Supplementary Table 1).

Rhizomes were mechanically harvested from each plot, with soil and roots carefully removed in the field before weighing. The commercial yield of rhizomes (t/ha) was calculated based on the recorded weights. Ginger variety one yielded 60 t/ha, while ginger variety two yielded 90 t/ha.

### Microbial community DNA extraction and high-throughput sequencing
All laboratory procedures were conducted at Shanghai Majorbio Bio-pharm Technology Co., Ltd. under standardised experimental conditions. Microbial DNA was extracted from 0.5 g of rhizosphere and bulk soil samples or 5 g of plant tissues utilising the DNeasy PowerSoil Kit (Qiagen, MD, USA), following the manufacturer's protocol. The quality of the extracted DNA was confirmed using 1% agarose gel electrophoresis, while concentration and purity were determined via a NanoDrop spectrophotometer (Thermo Fisher Scientific, USA). Each sample underwent analysis in triplicate and was stored at −20 °C until further processing. or bacterial community profiling, the V5–V7 hypervariable regions of the 16S rRNA gene were amplified using the universal primers 799 F (5′-AACMGGATTAGA-TACCCKG-3′) and 1193R (5′-ACGTCATCCCTACCTTCC-3′). For the fungal communities, amplification targeted the ITS2 region, employing primers ITS3F (5′-GCATCGATGAAGAACGCAGC-3′) and ITS4R (5′-TCCTCCGCTTATTGATATGC-3′).

The PCR amplification products were visualised on agarose gels, excised, and then purified with the AxyPrep DNA Gel Recovery Kit (Axygen, USA) according to the manufacturer's instructions. Purified amplicons were quantified using the Quantifluor™-ST Blue Fluorescence System (Promega Corporation, Madison, WI, USA). Equimolar quantities of replicate PCR products from the same sample were pooled for subsequent library preparation. Amplicon libraries were constructed following the Illumina metagenomic library preparation guidelines, with sequencing performed on the Illumina platform as detailed below.

### Amplicon Sequencing and Bioinformatic Analysis
Amplicon sequences for bacterial and fungal communities from thirty-six samples were sequenced independently on the Illumina MiSeq platform, generating paired-end reads of 250 bp. Negative controls were included to rule out contamination. Sequences were analysed using the Majorbio Cloud Platform[17] and the QIIME 1.9.1 pipeline[18]. A total of 2,537,316 high-quality reads were assembled, and low-quality reads were trimmed using USEARCH v.11.0[19]. Operational taxonomic units (OTUs) were assigned at a 97% similarity threshold and checked for chimaeras using the UPARSE pipeline[20]. Taxonomic annotation was performed using the SILVA database for bacteria and the UNITE database for fungi. Single rarefaction was performed at the shallowest sample depth, resulting in a subsampled dataset of 36,633 bacterial and 33,848 fungal sequences.

## Microbiome assembly

The neutral community model (NCM) was used to evaluate species composition[21]. The mean nearest-taxon distance (βMNTD) metric was calculated to assess community phylogenetic turnover. A null modelling approach with 999 randomisations was used to compute the βNTI index, which measures the divergence between observed and expected βMNTD values. The RCBray index was employed alongside a null model of the Bray–Curtis β diversity index to enhance βNTI assessment[5]. The normalised stochasticity ratio (NST) was calculated to quantify the contributions of deterministic and stochastic processes in community assembly[22].

## Predictive and statistical analysis

Data are presented as averages from at least three independent replicates with standard deviations. Microbial diversity was assessed using observed richness (Sobs) and the Shannon H index. Pairwise comparisons were performed using the Kruskal–Wallis test. Principal coordinate analysis (PCoA) was conducted using the vegan package v.2.4.3 in R software v. 3.3[23]. Permutational multivariate analysis of variance (PERMANOVA) based on Bray–Curtis dissimilarities was performed with 999 permutations ($n = 36$) using the vegan package in R. Fixed factors included plant compartment (bulk soil, rhizosphere, root, rhizome, stem, leaf) and variety (high-yield vs. low-yield). In addition, interaction terms between compartment × variety were tested. No random factors were included in the model. Core and specialist taxa were identified, and statistical analyses were conducted using R (v4.2.1). The FAPROTAX tool was used to infer microbial functional assemblages[24], and fungal OTUs were classified using FUNGuild[25]. Linear discriminant analysis effect size (LEfSe) was used to identify significant features[26]. Co-occurrence networks (SparCC) and keystone taxa identification followed MENAP protocols. Distance-based redundancy analysis (db-RDA) was performed to analyse relationships between microbial genera and environmental factors. Differential abundance (LEfSe), db-RDA, and correlations (Pearson/Mantel tests) were FDR-corrected. Distance-based redundancy analysis (db-RDA) was conducted separately for bacterial and fungal communities using the top 25 genera to assess the influence of soil nutrient variables. Nutrient parameters (total N, total P, total K, available P, and available K) were treated as continuous explanatory variables. Multi-collinearity was assessed by the variance inflation factor (VIF), and variables with a VIF > 10 were excluded from the final models.

## Metabolomics analysis

An untargeted metabolomics approach was used to investigate metabolic changes associated with the endophytic microbiome of ginger plants and their impact on plant yield. We analysed twenty-four samples, including leaves, stems, roots, and rhizomes from plants of both varieties. About 50 mg of each tissue sample was placed in a 2 ml centrifuge tube with a 6 mm grinding bead. Metabolite extraction involved 400 µl of methanol: water (4:1, v/v) containing 0.02 mg/ml l-2-chlorophenylalanine as an internal standard. Tissue disruption was performed using a Wonbio-96C frozen tissue grinder (Shanghai Wanbo Biotechnology Co., Ltd.) for 6 min at –10 °C and 50 Hz, followed by ultrasonic extraction at 5 °C for 30 min (40 kHz). The extracts were incubated at –20 °C for 30 min, then centrifuged at 13,000 g for 15 min at 4 °c. The supernatant was transferred to autosampler vials for LC-MS/MS analysis.

Pooled quality control (QC) samples were prepared by mixing equal aliquots of all extracts and processed in the same way to assess instrument stability. Untargeted metabolomic profiling was performed at Majorbio Bio-Pharm Technology Co., Ltd. (Shanghai, China) using a Sciex UPLC-Triple TOF™ 5600+ system coupled with an Acquity HSS T3 column (100 mm × 2.1 mm, 1.8 µm; Waters, USA). Solvent A consisted of 0.1% formic acid in a water: acetonitrile (95:5, v/v) mixture, while solvent B contained 0.1% formic acid in acetonitrile:isopropanol: water (47.5:47.5:5, v/v/v). Chromatographic separation was performed at 40 °C with a flow rate of 0.40 ml/min. The UPLC system was connected to a quadrupole time-of-flight mass spectrometer equipped with an ESI source operating in both positive and negative ionisation modes. Optimised parameters included a source temperature of 550 °C, curtain gas set at 30 psi, ion source gases 1 and 2 at 50 psi each, ion spray voltage floating at –4000 V (negative mode) and 5000 V (positive mode), declustering potential of 80 V, and collision energy ranging from 20 to 60 eV for MS/MS.

Data acquisition was performed in Information Dependent Acquisition (IDA) mode across a mass range of 50–1000 m/z. Raw data processing involved Progenesis QI (Waters Corporation, USA), resulting in a three-dimensional data matrix comprising features, retention times, and intensities. Internal standard peaks and known artefacts (noise, column bleed, reagent peaks) were removed. Metabolite annotation used HMDB, Metlin (https://metlin.scripps.edu/), KEGG, and the Majorbio in-house library. Only features detected in at least 80% of samples from one or more groups were retained, with peak intensities normalised using sum normalisation. Dataset quality was verified through relative standard deviation (RSD < 0.3) and a peak detection rate exceeding 70%.

Multivariate and univariate statistical analyses were performed on the Majorbio cloud platform (cloud.majorbio.com). Orthogonal Projections to Latent Structures Discriminant Analysis (PLS-DA) was carried out using ropls v.1.6.2 (R software), identifying metabolites with variable importance in projection (VIP) ≥ one and p < 0.001 as significantly different. Additionally, Pearson's correlation based on Bray-Curtis distance assessed associations between microbial genus-level abundances and metabolite profiles, while Procrustes analysis evaluated congruence between metabolomic and microbiome PCA ordinations. Factors examined in the multivariate analysis included varieties with different productivity levels, tissue types (leaf, stem, root, rhizome), and their interactions.

## In vitro metabolite addition assay

To assess the influence of microbiome-associated signalling metabolites on microbial growth dynamics, in vitro assays were conducted using two keystone taxa identified from co-occurrence network analyses: *Devosia riboflavina* and *Talaromyces pseudofuniculosus*. Two representative metabolites were selected: 1-oleoyl-lysophosphatidic acid (LPA; purity ≥99.89%, CAS 65528-98-5; MCE, Cat. No. RH773618) and oxindole (purity ≥99.89%, CAS 59-48-3; AmBeed, Cat. No. A136297), based on their positive (LPA) or negative (oxindole) correlation with beneficial microbes involved in nitrogen cycling and plant development. Metabolite stock solutions (10 mg/mL) were prepared in absolute ethanol and added to media at final concentrations of 0 µM (ethanol control), 25 µM, and 50 µM. Ethanol concentrations in all treatment groups were normalised to the ethanol control group, with a final concentration of 0.47%.

## Microbial strains and culture conditions

*Devosia riboflavina* (CGMCC 1.10398) was cultured in 8 mL of Reasoner's 2A (R2A) broth[27] (Sigma-Aldrich) in sterile 50 mL polypropylene tubes at 30 °C with shaking at 200 rpm. Treatments included: R2A alone (control), R2A + 25 µM LPA, R2A + 50 µM LPA, R2A + 25 µM oxindole, R2A + 50 µM oxindole. Each treatment was performed in six biological replicates. An autochthonous strain of *Talaromyces pseudofuniculosus* was grown in 3 mL of R2A broth in similar tubes at 28 °C, under static conditions, and subjected to the same metabolite treatments. All procedures were conducted aseptically.

## Growth monitoring and biomass quantification

For bacterial cultures, 1 mL aliquots were collected at 0, 24, 48, and 72 h. Optical density at 600 nm ($OD_{600}$) was measured using a UV-Vis spectrophotometer (Thermo Scientific Genesys 20) and converted to estimated colony-forming units (CFU/mL) using a standard calibration curve (1 $OD_{600} \approx 1 \times 10^9$ CFU/mL). Fungal biomass was quantified using dry weight. Cultures were vacuum-filtered through pre-weighed Whatman No. 1 filter paper, washed with sterile distilled water, dried at 60 °C for 48 h, and reweighed. Biomass was recorded in milligrams. Six biological replicates were included per treatment.

### Plant growth promotion assay

Tissue culture-derived ginger plantlets ($n = 6$ per treatment) were transplanted into 0.96-litre pots containing a sterilised substrate mixture of cocopeat, vermiculite, and perlite (1:1:1, v/v/v). Plants were cultivated under controlled chamber conditions: 12 h light/dark photoperiod, light intensity of 46 μmol/m²/s, 75% relative humidity, and 26 °C. Bottom-watering was used to minimise contamination. Four treatment groups were evaluated: Mock-inoculated control, *D. riboflavina* ($1 \times 10^8$ CFU/mL), *T. pseudofuniculosus* ($1 \times 10^8$ spores/mL), and Microbial consortium (mix of both strains at the same concentrations). Each microbial suspension (1 mL) was applied directly to the root zone during transplanting.

### Validation experiment data collection and statistical analysis

Bacterial growth data were modelled using a logistic growth function fitted by nonlinear regression (Python SciPy curve_fit), yielding estimates for the growth rate (r), carrying capacity (K), and inflexion point ($t_0$). Fungal growth was analysed via dry weight at each time point. Plant phenotypic traits were measured at 0, 15, and 21 days post-inoculation (dpi), including height (cm), leaf count, and shoot number. The following growth metrics were derived: Absolute Growth Rate (AGR, cm/day) = (Height$_{(21\ dpi)}$ − Height$_{(0\ dpi)}$ /21. Relative Growth Rate (RGR, 1/day) = ln(Height$_{(21\ dpi)}$/ Height$_{(0\ dpi)}$)/21. Δ Leaves = Leaves$_{(21\ dpi)}$–Leaves$_{(0\ dpi)}$. Δ Shoots = Shoots$_{(21\ dpi)}$ − Shoots$_{(0\ dpi)}$. A composite growth index was calculated as the sum of these three traits. The Composite Growth Index at 30 dpi was predicted by linear regression of composite scores across 0, 15, and 21 dpi. Statistical analyses were performed in Python and GraphPad Prism 9. One-way ANOVA with Tukey's post hoc test was used to compare treatment effects. Pairwise comparisons at individual time points were conducted using Student's t-tests. Differences were considered statistically significant at $p < 0.05$.

### Statistics and Reproducibility

All statistical analyses were conducted using the Majorbio cloud platform (cloud.majorbio.com) and software tools, specifically R (version 4.2.0, utilizing the ropls package v1.6.2 for partial least squares discriminant analysis) and Python SciPy (version 1.0.0). *P*-values were adjusted for multiple comparisons using the Benjamini–Hochberg false discovery rate (FDR) method, maintaining a significance threshold of $p < 0.05$ (adjusted) unless otherwise specified.

In our multivariate analyses, we incorporated experimental variables, including plant variety (differentiating based on productivity), tissue type (leaf, stem, root, rhizome), and soil type (bulk vs. rhizosphere), along with their interactions. Pearson's correlation and Procrustes analyses explored the associations between microbiome and metabolome data sets.

Biological replicates consisted of independent samples from distinct plants in their agronomical habitats. We assessed 36 biological samples, encompassing bulk soil, rhizosphere soil, leaves, stems, roots, and rhizomes from two varieties of ginger for metagenomic and metabolomic profiling. Technical replicates—about DNA quantification, PCR amplification, and LC–MS/MS injections—were conducted to ensure protocol fidelity. However, subsequent statistical evaluations did not consider these technical replicates as independent biological samples.

To address reproducibility, pooled quality control (QC) samples were analysed throughout the metabolomics workflow to monitor instrument stability, achieving a relative standard deviation (RSD) of under 0.3%. Moreover, all sequencing and metabolomic experiments were repeated across at least two independent batches, yielding consistent microbial community structure and metabolite profiles. Non-quantitative observations, such as representative images of the plants, were reproduced across various specimens, with selected examples presented in the figures.

## Results

### Overview of sequencing and De Novo assembly

We analysed 36 composite samples from two cultivars with contrasting yields to profile the ginger-associated microbiomes. High-throughput sequencing generated 1.32 million quality-filtered reads for bacteria/archaea and 1.22 million for fungi, identifying 5970 bacterial/archaeal OTUs and 1701 fungal OTUs. Bacterial communities spanned 46 phyla, while fungal taxa were classified into 17 phyla, as annotated against the SILVA and UNITE databases (Supplementary Data 1). Rarefaction curves (Supplementary Fig. 1) and Shannon diversity indices (Supplementary Fig. 2) confirmed that sequencing depth was sufficient to capture most community diversity. Bacterial richness was more than threefold higher than fungi's, highlighting their broader taxonomic diversity across soil and plant compartments.

### Contrasting assembly mechanisms in bacterial and fungal communities

We applied the neutral community model to elucidate the mechanisms driving microbial community assembly in two ginger varieties with distinct yield characteristics. Our analysis revealed contrasting assembly patterns between bacterial and fungal communities. Bacterial communities were predominantly shaped by stochastic processes, as evidenced by strong fits to the neutral model ($R^2 = 0.67$ and 0.68) and high migration rates (Nm = 1353 and 1440) for both cultivars (Fig. 1a). In contrast, fungal communities exhibited poor fits to the neutral model ($R^2 = -0.03$ and $-0.08$), indicating a more substantial influence of deterministic processes (Fig. 1b). To dissect these assembly mechanisms further, we calculated the β-nearest-taxon index (βNTI), which assesses niche-level assembly processes. Bacterial communities displayed elevated mean βNTI values in both cultivars' rhizosphere and root tissues, pointing to a dominant deterministic influence in these niches (Fig. 1c). Conversely, fungal communities showed evidence of deterministic assembly, with strong homogeneous selection (βNTI < -2) across various plant tissues (Fig. 1d).

Compositional analyses revealed significant niche-specific variations in bacterial communities (Fig. 1e). In the first variety's bulk soil, community structure was mainly shaped by dispersal limitation (66.67%). In contrast, in the second variety, heterogeneous selection was more dominant (44.44%). Both varieties exhibited substantial drift effects in stem and leaf samples, highlighting the diversity of assembly processes across niches. Fungal community assembly, however, followed a different trajectory. While stochastic processes dominated in bulk and rhizosphere soils (100%), root tissues displayed a mix of influences. The roots of the first variety were primarily shaped by drift (77.78%), whereas those of the second variety were influenced by a combination of drift, homogeneous dispersal, and selection (Fig. 1f).

We used the normalised stochasticity ratio (NST) to evaluate the balance between stochastic and deterministic assembly, with 50% as the threshold. Bacterial endophytic communities exhibited considerable stochasticity (NSTavg > 0.5), particularly in the rhizomes and stems of the first variety (Fig. 1g). In contrast, fungal communities in both cultivars were predominantly governed by deterministic processes (Fig. 1h).

Distinct ecological processes shaped bacterial communities in different plant compartments. Deterministic selection dominated the rhizosphere and endosphere, while stochastic influences were more evident in the bulk soil. Fungal communities in roots shifted from drift in the low-yield variety to mixed deterministic processes in the high-yield counterpart.

### Host genotype modulates niche-specific microbial structuring

Because the two varieties were grown in different locations, standardised cultivation practices and synchronised sampling were applied to reduce potential environmental variation. This design allows observed microbiome and metabolome differences to be more confidently attributed to host genotype rather than site-specific management effects.

A comprehensive analysis revealed that microbial communities in ginger are strongly compartmentalised and influenced by host genotype. Kruskal–Wallis test indicated a significant reduction in alpha diversity ($P < 0.05$) in aerial tissues relative to soil and subterranean compartments across both varieties (Fig. 2a, b), underscoring the importance of the root-soil interface in sustaining microbial richness—likely due to its greater

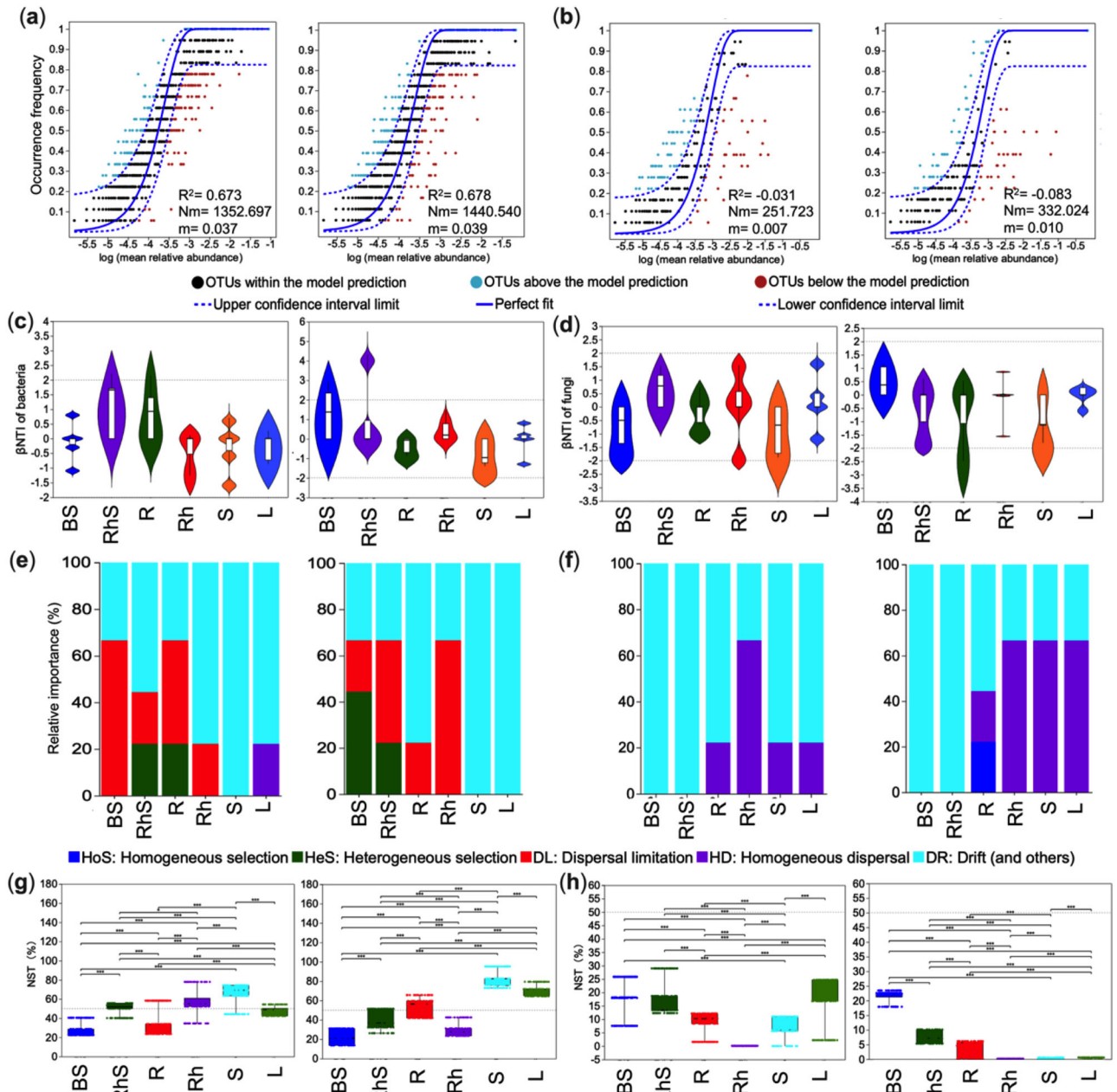

**Fig. 1 | Ecological processes governing the assembly of bacterial and fungal communities across microbial niches in two ginger cultivars with contrasting yields.** This figure illustrates the relative influence of stochastic versus deterministic forces on community assembly for bacterial and fungal microbiomes. Neutral community model (NCM) fits for bacterial (**a**) and fungal (**b**) communities. The solid blue line indicates the Sloan model fit, with dashed lines denoting the 95% confidence intervals. Operational taxonomic units (OTUs) falling significantly above or below the model are shown in green and red, respectively. Model performance is summarised by the coefficient of determination ($R^2$) and the product of metacommunity size and immigration rate (Nm). Violin plots of the β-nearest taxon index (βNTI) for bacterial (**c**) and fungal (**d**) communities. Values of |βNTI|≥ 2 indicate deterministic assembly, while |βNTI|< 2 suggests stochasticity. Relative contributions of distinct ecological processes to bacterial (**e**) and fungal (**f**) community assembly, inferred from βNTI and Bray–Curtis–based Raup–Crick index (RCBray). Normalised stochasticity ratio (NST) for bacterial (**g**) and fungal (**h**) communities across compartments. The NST threshold of 0.5 (dashed line) separates predominantly deterministic (<0.5) from stochastic (>0.5) dynamics. BS bulk soil, RhS rhizosphere soil, R root, Rh rhizome, S stem, L leaf. Panels are split by cultivar: variety one (left) and variety two (right).

nutrient heterogeneity and physical complexity. Principal coordinate analysis (PCoA) based on Bray–Curtis dissimilarity confirmed clear niche segregation in bacterial communities, which was statistically supported by Adonis testing ($R^2$ = 0.7241 for variety one; $R^2$ = 0.6908 for variety two; $P$ = 0.001) (Fig. 2c).

Notably, the endophytic communities associated with the underground environment of variety two exhibited more significant dissimilarity than those of variety one (Fig. 2c), suggesting varietal-specific influences on microbial assembly. Microbial composition analysis, visualised through floral and bar graphs, identified generalist and niche-specific microbial taxa. The high-yield variety hosted 774 bacterial genera in bulk soil (vs. 746 in low-yield) and enriched nitrogen-fixing *Devosia* (see Fig. 2d–e and Supplementary Data 2). Fungal communities were niche-specialised, with only 0.64% generalists (Fig. 2f–h). Dominant fungal genera were highly adapted to the unique soil microbiomes of each variety, suggesting limited potential for direct colonisation of host tissues. This finding underscores the role of

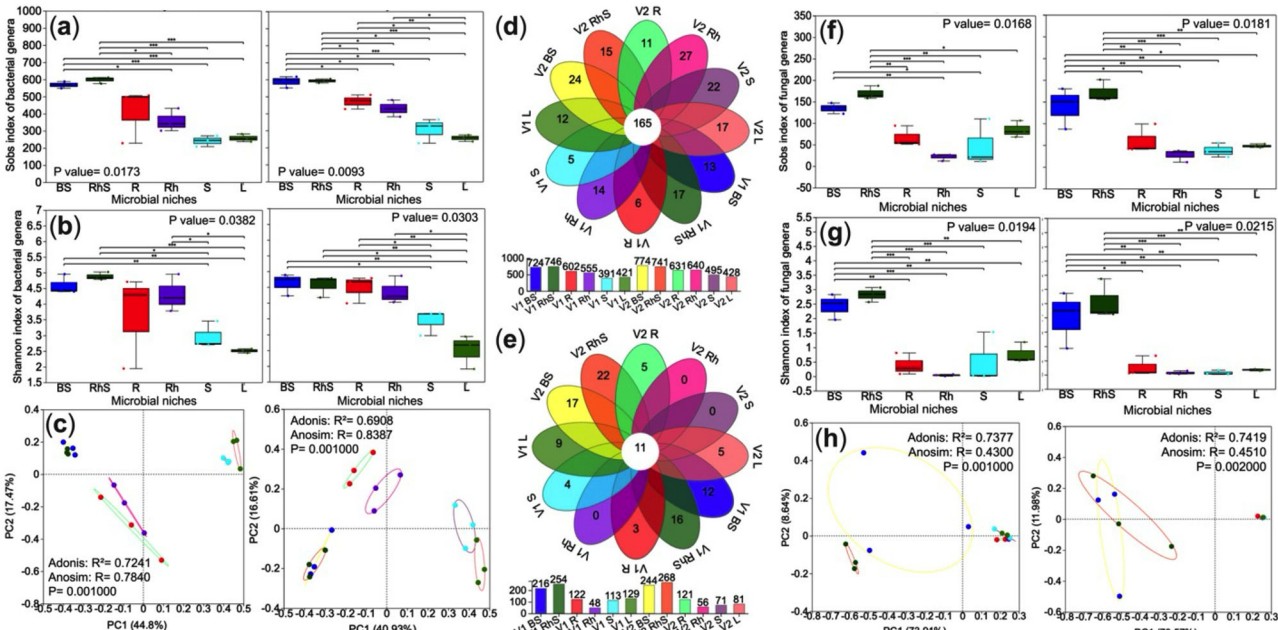

**Fig. 2 | Alpha and beta diversity of bacterial and fungal communities across microbial niches in two ginger varieties differing in yield.** This figure summarises microbial diversity patterns and niche-specific taxa in bulk soil, rhizosphere, and internal tissues. **a, b** Alpha diversity of bacterial communities based on the observed species richness (Sobs) and Shannon index. **c** Principal coordinate analysis (PCoA) of bacterial beta diversity using Bray–Curtis distances, with statistical significance assessed by Adonis/PERMANOVA. Venn diagrams showing shared (central ring) and unique bacterial (**d**) and fungal (**e**) genera across all compartments, with bar charts indicating the total number of genera per niche. Details on niche-specific genera are provided in Supplementary Data. Fungal alpha diversity was evaluated using Sobs (**f**) and Shannon (**g**) indices. **h** PCoA of fungal beta diversity with significance tested by Adonis/PERMANOVA. All comparisons are presented in the figures, and comprehensive statistical results are available in Supplementary Data. *Statistical differences* were assessed using the Kruskal–Wallis test with Benjamini–Hochberg correction; significance is denoted as $p < 0.05$ (*), $p < 0.01$ (**), and $p < 0.001$ (***). *Sample type* BS bulk soil, RhS rhizosphere soil, R root, Rh rhizome, S stem, L leaf. Panels are grouped by cultivar: variety one (left) and variety two (right).

the root system as a selective interface, filtering and shaping microbial communities. The distinct separation between soilborne and endophytic communities (Fig. 2f, g) was further supported by unique beta diversity patterns (Fig. 2h), emphasising the strong influence of the root system on microbial assembly processes.

**Host metabolites shape microbiome composition and function**

To investigate how host metabolites influence microbiome assembly and yield outcomes, we performed untargeted LC–MS/MS profiling across ginger tissues from two cultivars. A total of 586 unique metabolites were identified, with 511 matched to spectral libraries and 200 annotated in KEGG pathways. Principal component analysis (PCA) revealed clear separation of metabolite profiles according to both plant tissue type and cultivar yield level (Fig. 3a). Samples from the high-yield cultivar clustered distinctly from the low-yield cultivar along PC1, which explained 57.1% of the total variance, indicating cultivar-specific metabolic reprogramming. PC2 (27.4% variance) primarily separated tissues, with rhizome and root samples forming tighter clusters relative to leaf and stem samples. These results suggest that genotype and tissue identity shape the ginger metabolome, with yield-associated varieties exhibiting coordinated shifts in primary and secondary metabolite production.

KEGG pathway enrichment analysis revealed that several biosynthetic and metabolic pathways were significantly upregulated in the high-yield ginger cultivar (Fig. 3b). Notably, isoflavonoid biosynthesis and folate metabolism were among the most enriched, both of which are implicated in plant defence and growth. Additional pathways included fructose and mannose metabolism, amino sugar metabolism, and glycerophospholipid metabolism, suggesting broad-scale carbon and nitrogen flow reprogramming. Enriching secondary metabolite biosynthesis pathways aligns with the observed increase in yield-associated microbial taxa, supporting a

potential role for plant-derived compounds in shaping beneficial microbiome composition.

Differential metabolite analysis identified several compounds with significantly higher abundance in the high-yield cultivar (VIP > 1, FDR < 0.05; Fig. 3c). Among these, Niazimin A (an alkaloid), 1-oleoyl lysophosphatidic acid (a lipid mediator), and PA (16:0/18:2) were particularly enriched in rhizome and root tissues. Goshuyic acid also showed consistent upregulation across tissues. In contrast, oxindole was more abundant in the low-yield cultivar and may be linked to microbial or metabolic signatures of suboptimal performance. The identity and distribution of these metabolites suggest that they serve as biochemical signals, influencing the microbiome assembly, nutrient exchange, or host defence. To evaluate the ecological roles of these compounds, we analysed the correlations between metabolite concentrations and microbial abundances in various tissues. In the high-yielding cultivar, 1-oleoyl lysophosphatidic acid was strongly associated with *Devosia* and Subgroup_10 (r > 0.75, P < 0.01), which are involved in nitrogen fixation. Niazimin A correlated with *Sphingomonas*, while (S)-5-amino-3-oxohexanoate was positively associated with *Flavobacterium* and *Devosia*. Conversely, the metabolite oxindole—more abundant in the low-yield cultivar—showed negative correlations with these same taxa (Fig. 3d).

Fungal community structure also mirrored these metabolite patterns. In the low-yield cultivar, (S)-5-amino-3-oxohexanoate positively correlated with *Curvularia*, *Mortierella*, and *Pseudaleuria*, while *Apiotrichum* showed negative associations with P-menth-8-en-3-ol. In the high-yielding cultivar, *Pichia* and *Sporidiobolus* were negatively associated with 1-oleoyl lysophosphatidic acid and aminoacetone (Fig. 3e).

These findings indicate that metabolites produced by the host serve as selective filters, influencing the diversity and activity of bacterial and fungal taxa associated with nutrient cycling, disease suppression, and the plant's overall performance.

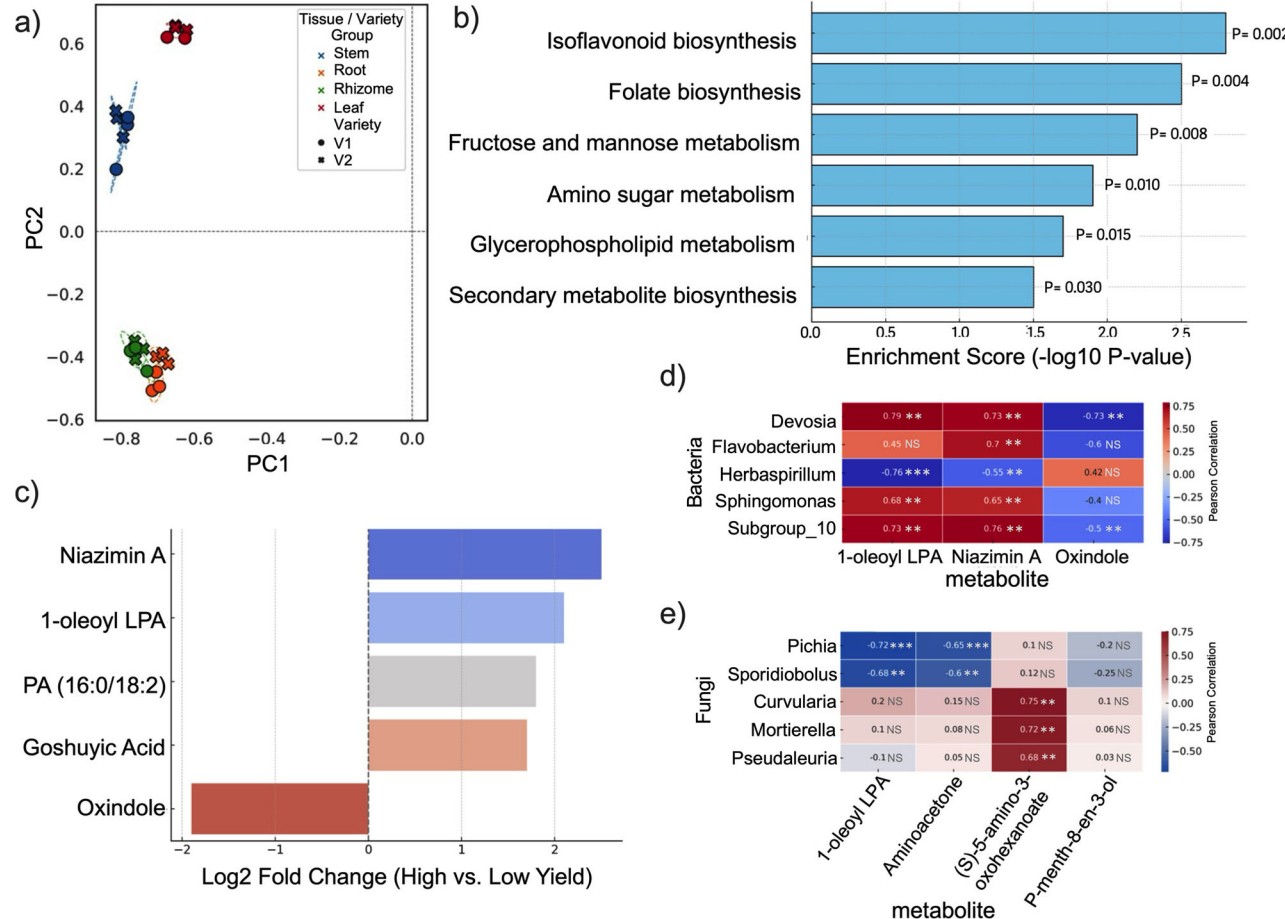

**Fig. 3 | Host metabolite profiles and microbe–metabolite associations in ginger cultivars with contrasting yields. a** Principal component analysis (PCA) of untargeted LC-MS/MS metabolite profiles, showing separation by tissue type and cultivar (V1: low yield; V2: high yield). **b** KEGG pathway enrichment in high-yield tissues reveals significant upregulation of isoflavonoid and folate biosynthesis, among other metabolic pathways. **c** Differentially abundant metabolites between high- and low-yield cultivars, including Niazimin A, 1-oleoyl lysophosphatidic acid (LPA), and Oxindole. **d** The Pearson correlation heatmap shows strong positive associations between host metabolites and growth-promoting bacteria (e.g., *Devosia*) and negative associations with Oxindole. **e** The correlation heatmap of key host metabolites and fungal taxa indicates differential microbial responses and potential host-driven filtering. Numbers in each cell indicate Pearson correlation coefficients. Statistical significance is indicated by asterisks (*$p < 0.05$, **$p < 0.01$, ***$p < 0.001$).

## Keystone Taxa and network complexity underlie yield differences

We focused on dominant operational taxonomic units (OTUs) with a relative abundance of ≥2% to identify keystone microbial taxa within the ginger microbiome. We pinpointed potential biomarker taxa using linear discriminant analysis effect size (LEfSe) and differential abundance analyses. Network analysis further revealed keystone taxa based on their topological roles:

Bacterial nodes were classified as network hubs if they exhibited a degree >13 and a closeness centrality >0.08, while fungal nodes required a degree >42 and a closeness centrality >0.22. These criteria identified taxa with central roles in microbial networks, suggesting their potential as keystone representatives within the ginger microbiome.

In this study, *hub* and *keystone taxa* are related but not synonymous. We define hub taxa based on network topology — specifically, nodes with high degree and closeness centrality values, indicating structural centrality within co-occurrence networks. In contrast, keystone taxa refer to microbial taxa with a disproportionately large impact on community structure or function, as supported by multi-omics correlations, ecological relevance, or enrichment in high-yield systems. While some taxa may fulfil both criteria, we use these terms with context-specific precision to distinguish between structural centrality and ecological function.

## Taxonomic and functional differentiation of ginger-associated microbial communities

Comparative analysis of microbial communities in two ginger varieties revealed striking taxonomic and functional divergences that correlate with yield variation and niche adaptation.

Genus-level resolution identified *Sphingomonas*, *Methylobacterium-Methylorubrum*, and *Bacillus* as core taxa across both varieties. Notably, *Flavobacterium* was enriched in variety one (4.73%), whereas *Acidovorax* was more prevalent in variety two (3.14%). Tissue-specific patterns highlighted clear microbial compartmentalisation: *Sphingomonas* dominated leaf tissues, while *Bacillus* was most abundant in bulk soil. The nitrogen-fixing *Allorhizobium-Neorhizobium-Pararhizobium-Rhizobium* complex exhibited niche preference for stems in variety one and roots/rhizomes in variety two. Remarkably, *Flavobacterium* reached 22.91% abundance in the roots of variety one (Fig. 4a).

Fungal communities were primarily composed of unclassified taxa (>68% in both varieties), with Ascomycota as the dominant phylum. *Pseudaleuria* and *Lophotrichus* emerged as prevalent genera (Fig. 4b). LEfSe analysis further identified variety-specific fungal biomarkers such as *Gymnoascus*, *Torula*, and *Powellomyces* for variety one and *Talaromyces* for variety two, highlighting divergent evolutionary trajectories in rhizosphere adaptation (Fig. 4c). Microbial biomarker analysis revealed 24

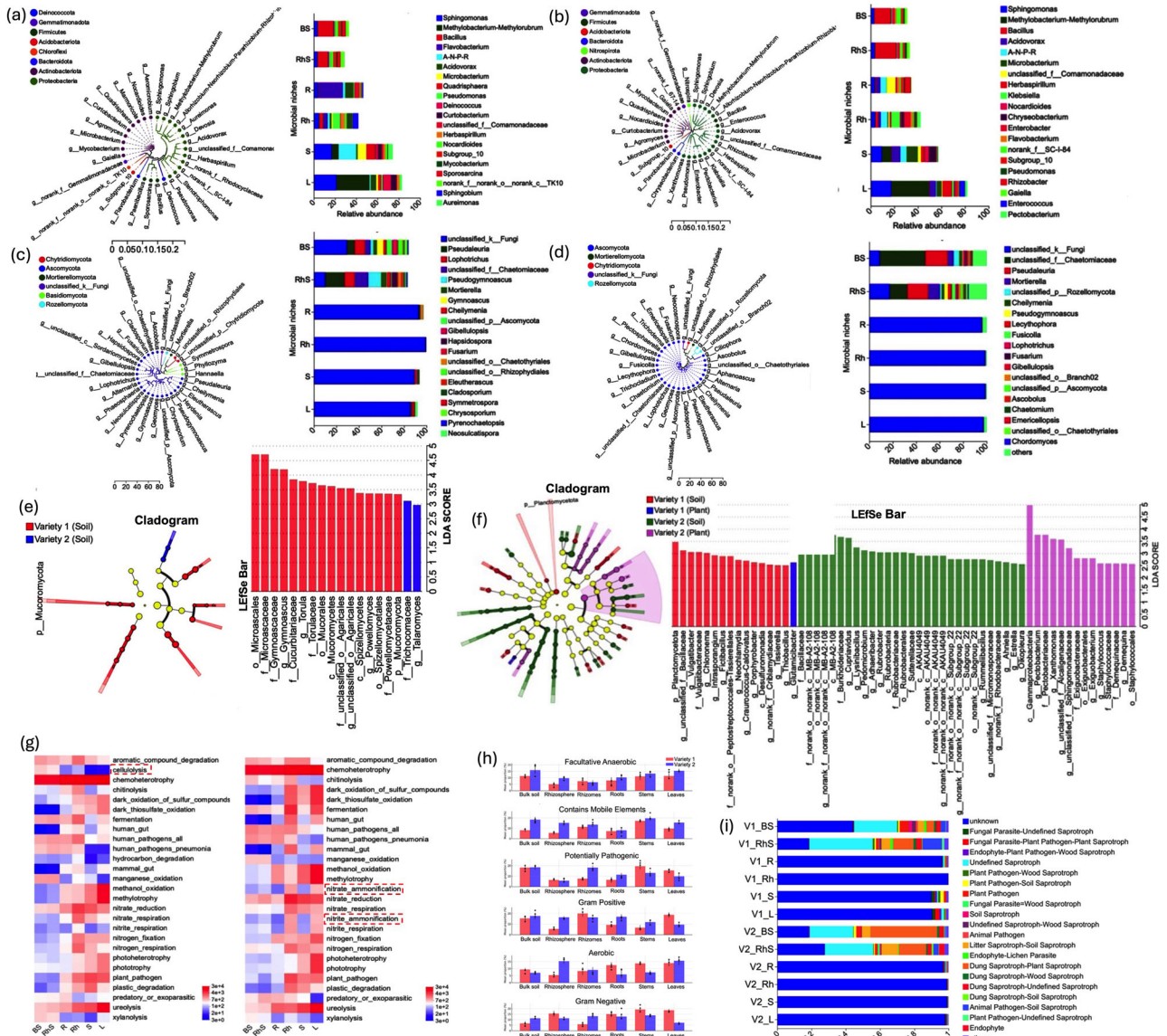

**Fig. 4 | Composition, biomarkers, and functional profiles of bacterial and fungal communities across ginger varieties and compartments.** This multi-panel figure integrates taxonomic composition, indicator taxa, and predicted ecological functions of bacterial and fungal communities associated with two ginger varieties. **a, b** Relative abundance of the top 20 bacterial genera across tissues and varieties, visualized as stacked bar plots. **c, d** The relative abundance of the top 20 fungal genera is structured similarly (**a, c** V1: low yield; **b, d**, V2: high yield). Discriminatory taxa identified by LEfSe analysis (LDA > 2.5, P < 0.05), showing bacterial (**e**) and fungal (**f**) biomarkers between varieties, displayed as LDA score plots and clado-grams. **g** Functional prediction of bacterial communities based on FAPROTAX, highlighting nitrogen-related processes such as nitrate/nitrite ammonification and nitrate reduction. **h** BugBase-derived prediction of bacterial functional groups. Bars show mean ± SD. Individual points represent biologically independent replicates (n = 3). **i** Fungal guild classification of fungal communities via FUNGuild, distinguishing dominant trophic strategies across compartments. BS bulk soil, RhS rhizosphere soil, R root, Rh rhizome, S stem, L leaf, V1 variety one, V2 variety two.

discriminatory bacterial genera (LDA > 2.5, P < 0.05). Variety one was characterised by soilborne taxa such as *Vulgatibacter* and *Intrasporangium*. In contrast, variety two was enriched in *Cupriavidus*, *Pectobacterium*, and *Rubrobacter* (Fig. 4d). Despite these taxonomic differences, functional predictions using FAPROTAX suggested only modest divergence. Variety one showed higher cellulolytic activity, while variety two displayed enriched nitrogen metabolism, particularly in rhizome and foliar compartments, including ammonification and nitrate reduction pathways (Fig. 4e).

Phenotypic predictions using BugBase distinguished functional traits between niches. Variety two exhibited traits associated with facultative anaerobes, especially in rhizomes and stems (Fig. 4f). FUN-Guild analysis corroborated these distinctions, revealing differential representation of saprotrophs and pathogens between tissue types and varieties (Fig. 4g).

These findings highlight the significant influence of host genotype and environmental conditions on the composition and functional capabilities of microbial communities associated with ginger. By integrating taxonomic identification and metabolic analysis, we better understand how micro-biomes contribute to yield and resilience in ginger cultivation systems.

## Microbial cooccurrence networks and the identification of keystone microbial Taxa

Network analysis of the 50 most abundant microbial species revealed significant structural differences (Pearson r > 0.96; P < 0.05) between the two ginger varieties. In variety one, the bacterial microbiome was characterized by 163 nodes and 272 edges (see Fig. 5a). In comparison, variety two displayed a more complex network consisting of 155 nodes and 596 edges (see Fig. 5b). Notably, there were no negative interactions among the microbial

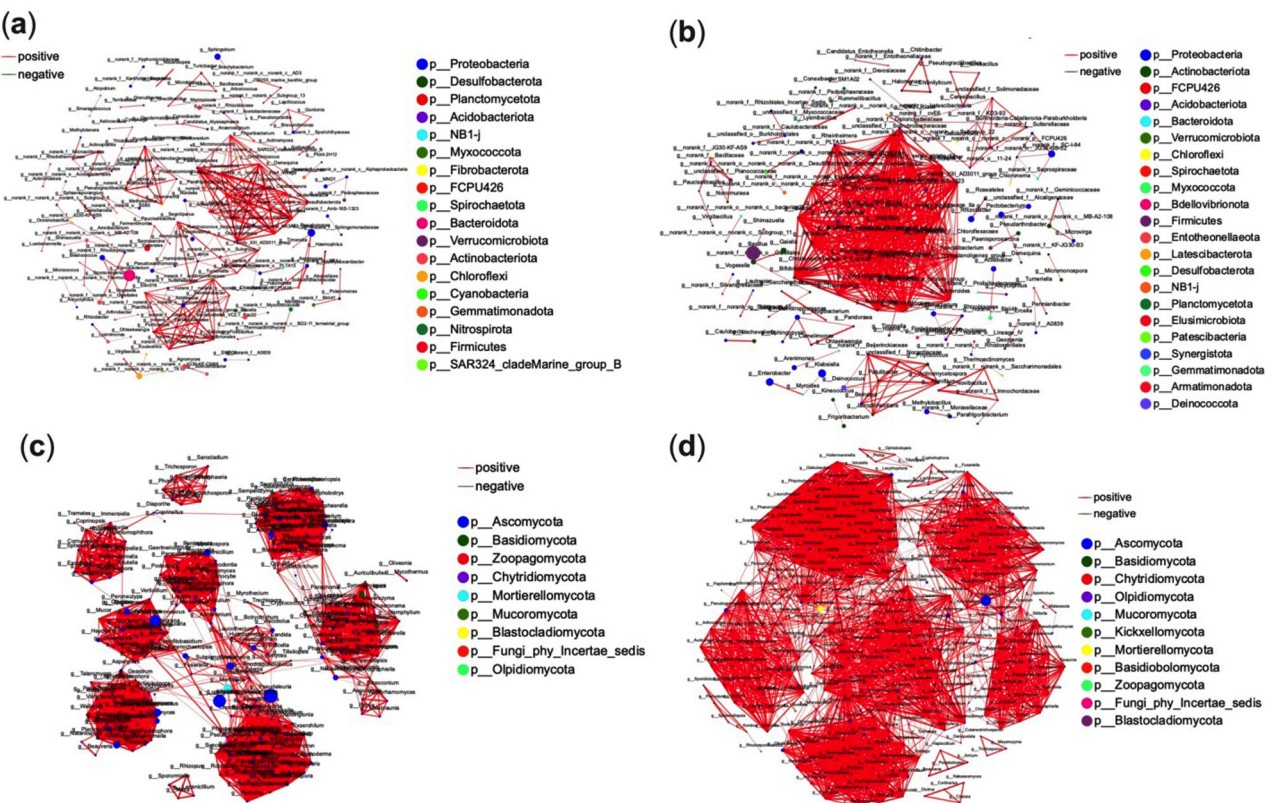

**Fig. 5 | Microbial co-occurrence networks and keystone taxa in ginger varieties with contrasting yields.** Co-occurrence networks illustrate the structural organisation of dominant bacterial (**a**, **b**) and fungal (**c**, **d**) genera in low-yield (**a**, **c**) and high-yield (**b**, **d**) ginger cultivars. Nodes represent genera, colour-coded by phylum, with node size proportional to degree (number of connections). Edges represent significant pairwise correlations (Pearson's $r > 0.96$, $P < 0.05$): red for positive, green for negative. Keystone taxa (network hubs) were defined by high degree centrality, indicating potential roles in stabilising or organising microbial community structure.

taxa in either ecosystem, highlighting a predominant trend of symbiotic relationships within these communities.

Variety two demonstrated greater complexity in bacterial interactions, with an edge-to-node ratio of 3.84 compared to 1.67 in variety one. The mean connectivity of variety two (7.65) also exceeded that of variety one (3.34), suggesting the presence of highly interconnected modules within its network. Furthermore, centralisation and density metrics were higher in variety two, reflecting a more connected and organised microbial structure. Eleven network hubs, including *Ulvibacter*, *Saccharothrix*, and *Paenalcaligenes*, were identified in variety one. In contrast, variety two harboured 27 network hubs, such as *Dialister*, *Lachnospira*, and *Fusicatenibacter*. These hub taxa represent keystone species critical for maintaining ecological integrity and functional stability.

Fungal co-occurrence networks also revealed greater complexity in variety two, with an edge-to-node ratio of 12.23 (Fig. 5d) compared to 11.04 in variety one (Fig. 5c). Keystone fungal taxa differed significantly between the varieties: variety one was characterised by *Myrmecridium*, *Pseudorobillarda*, *Duddingtonia*, and *Chalara*, while variety two featured *Microascus*, *Ganoderma*, *Talaromyces*, and *Leucothecium*, among others.

These findings underscore microbial communities' intricate and diverse nature, which is shaped by the ecological contexts of different ginger varieties. The discrepancies in keystone taxa highlight their potential influence on ecosystem dynamics, including nutrient interactions and microbial co-occurrence patterns. Further analyses will explore the functional roles of these keystone taxa and their contributions to the stability and resilience of ginger-associated microbial networks. Network analysis revealed the high-yield variety's microbiome was more interconnected, with 27 bacterial hubs (e.g., *Dialister*, *Lachnospira*) versus 11 in low-yield. Fungal

hubs (*Talaromyces*, *Ganoderma*) in the high-yield variety correlated with nutrient turnover.

## Nutrient dynamics mediate microbial recruitment

The relationships between nutrient dynamics and microbial community co-occurrences in ginger were investigated by analysing physicochemical parameters of soil and plant samples (see Supplementary Table 1). Multi-collinearity was assessed using the variance inflation factor (VIF) criterion, excluding factors with VIF > 10. In the first ginger variety, four parameters - total nitrogen (TN), total potassium (TK), total phosphorus (TP), and available potassium (AK) - were correlated with microbial communities. In contrast, the second variety included five parameters: TN, TK, TP, AK, and available phosphorus (AP). Distance-based redundancy analysis (db-RDA) was used to evaluate the influence of these nutrient factors on community structure. These variables explained approximately 20% of the variation in the assemblages of the 25 most prevalent bacterial (Fig. 6a) and fungal (Fig. 6b) genera. Significant genus-level associations were identified between nutrient parameters and distinct microbiomes across both varieties.

In the first variety, AK strongly correlated with the soil bacterial microbiome, TK with the rhizome microbiome, and TP and TN with the root microbiome (Fig. 6a, $P < 0.05$). AK and AP were associated with the leaf microbiome in the second variety, while TP, TN, and TK correlated with the soil microbiome (Fig. 6a, $P < 0.05$).

For fungal communities, AK correlated with the soil microbiome, and TP with the root and stem microbiomes in variety one (Fig. 6b, $P < 0.05$). In variety two, AK and AP were linked to the root, rhizome, stem, and leaf microbiomes, while TK and TN were associated with the rhizosphere microbiome, and TP with the bulk soil microbiome (Fig. 6b, $P < 0.05$). Co-

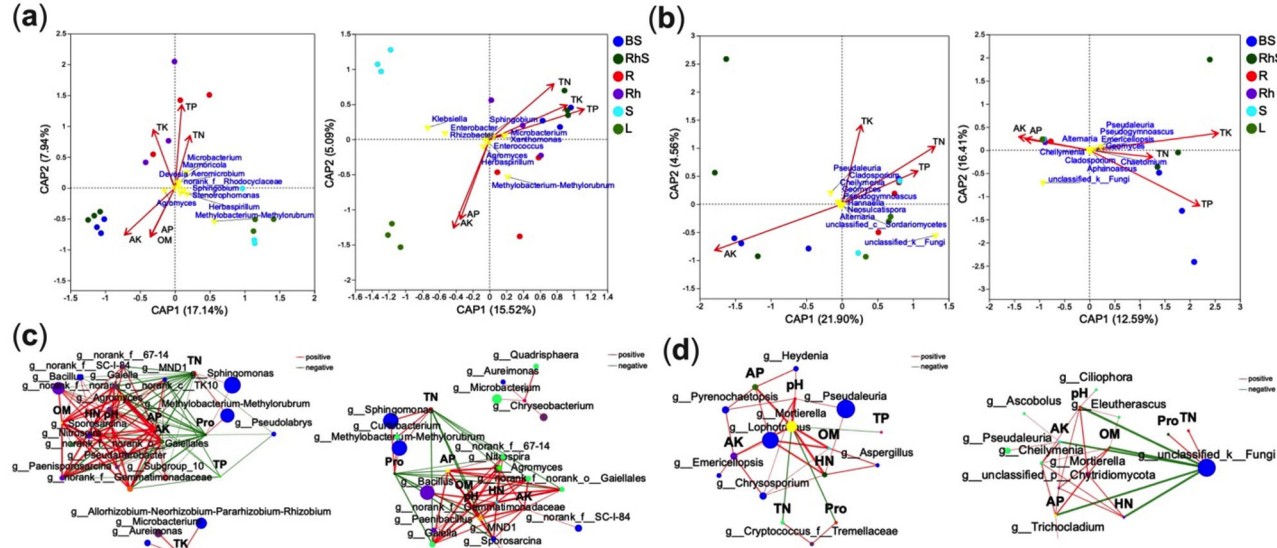

**Fig. 6 | Nutrient–microbiome interactions across ginger cultivars with contrasting yields.** Distance-based redundancy analysis (db-RDA) showing the influence of nutrient concentrations on bacterial (**a**) and fungal (**b**) community composition. Coloured points represent microbial niches, red arrows indicate nutrient vectors, and yellow labels denote responsive microbial taxa. Microbial–nutrient correlation networks for bacterial (**c**) and fungal (**d**) taxa. Nodes represent microbial genera, with size proportional to relative abundance and colour-coded by taxonomy. Edges represent significant correlations (r > 0.7, P < 0.05): red for positive, green for negative. Edge thickness reflects correlation strength. BS bulk soil, RhS rhizosphere soil, R root, Rh rhizome, S stem, L leaf. Variety one (left panels), variety two (right panels).

occurrence networks revealed significant correlations (Pearson's r ≥ 0.85, P < 0.05) between nutrient parameters and microbial taxa. The bacterial network comprised 38 nodes for both varieties, while the fungal network included 13 nodes for variety one and 19 for variety two.

In variety one, 27 bacterial taxa showed negative correlations with TN, while two were positively correlated. AK was positively associated with 22 taxa, TK with three, and TP negatively with 14. In variety two, TN negatively correlated with 20 genera and positively with three, while AK, AP, and TK positively associated with 15, 17, and five genera, respectively.

Key taxa were identified through node degree and closeness centrality analyses. For example, *Solirubrobacter* (r = 0.94), *Paenisporosarcina* (r = 0.93), and *Agromyces* (r = 0.93) positively correlated with AK (P < 0.01). *Microbacterium* (r = 0.84) and *Aureimonas* (r = 0.76) were positively associated with TK, while *Bacillus* (r = −0.71), *Nitrospira* (r = −0.74), and *Agromyces* (r = -0.76) negatively correlated with TN (P < 0.01). *Cryobacterium* (r = -0.70), *Solirubrobacter* (r = −0.71), and *Microvirga* (r = −0.72) were negatively correlated with TP (P < 0.01).

In variety two, *Agromyces* (r = 0.88), *Nitrospira* (r = 0.84), and *Bryobacter* (r = 0.82) positively correlated with AK (P < 0.01). *Quadrisphaera* (r = 0.81), *Klenkia* (r = 0.76), *Chryseobacterium* (r = 0.73), and *Aureimonas* (r = 0.73) were positively associated with TK (P < 0.01). *Curtobacterium* (r = 0.81) and *Methylobacterium-Methylorubrum* (r = 0.75) positively correlated with TN, while *Reyranella* (r = −0.72), *Agromyces* (r = −0.76), *Nitrospira* (r = −0.77), and *Solirubrobacter* (r = −0.79) showed negative correlations (P < 0.01). *Bryobacter* (r = −0.71) was negatively correlated with TP, while *MND1* (r = 0.92), *Nitrospira* (r = 0.91), *Gaiella* (r = 0.89), *Solirubrobacter* (r = 0.84), and *Agromyces* (r = 0.84) positively correlated with AP (P < 0.01) (Fig. 6d).

For fungal communities in variety one, *Mortierella* (r = 0.83), *Chrysosporium* (r = 0.79), *Pyrenochaetopsis* (r = 0.77), *Emericellopsis* (r = 0.75), and *Lophotrichus* (r = 0.70) positively correlated with AK (P < 0.01). *Mortierella* was negatively correlated with TN (r = −0.78) and TP (r = −0.72) (P < 0.01). In variety two, *Chrysosporium* (r = 0.91), *Penicillium* (r = 0.81), *Cheilymenia* (r = 0.79), and *Podospora* (r = 0.79) positively correlated with AP (P < 0.01). Nine taxa, including *Chrysosporium* (r = 0.84), *Cheilymenia* (r = 0.83), *Podospora* (r = 0.79), and *Penicillium* (r = 0.77), positively correlated with AK (P < 0.01). *Penicillium* (r = −0.70) was negatively correlated with TN (P < 0.01).

Total nitrogen and potassium explained 20% of microbial variation. *Agromyces* and *Solirubrobacter* correlated with available potassium, while *Mortierella* linked to TN/TP trade-offs. The high-yield variety exhibited stronger positive correlations with AP/AK, reflecting optimized nutrient-microbe synergy. This study highlights the complex interplay between nutrient dynamics and microbial communities in two ginger cultivars, providing valuable insights into plant-microbe interactions in agricultural ecosystems.

**Microbial Responses to LPA and Oxindole in In Vitro Assays**

The growth dynamics of *Devosia riboflavina* were significantly modulated by metabolite treatments. Under control conditions, the bacterium displayed a logistic growth pattern, reaching a maximum estimated biomass of 9.61 × 10⁸ CFU/mL. Supplementation with 1-oleoyl-LPA led to dose-dependent increases in final biomass (up to 1.03 × 10⁹ CFU/mL at 50 μM), albeit with modest reductions in growth rate. In contrast, Oxindole induced a biphasic response: 25 μM delayed growth onset ($t_0$ = 11.35 h) and reduced the rate, while 50 μM significantly enhanced proliferation speed (r = 0.2239 h⁻¹) but decreased final CFU (Fig. 7a).

*Talaromyces pseudofuniculosus* exhibited treatment-specific growth responses over the 72-hour incubation period (Fig. 7b). All groups began with comparable baseline dry weights (~1.0 mg at zero h). Biomass increased steadily in all treatments, but the magnitude and kinetics differed significantly. LPA supplementation notably stimulated fungal proliferation in a concentration-dependent manner. At 72 h, LPA 50 μM yielded the highest biomass (7.21 mg), considerably exceeding both the control (3.72 mg) and LPA 25 μM (4.78 mg) treatments (P < 0.05). In contrast, Oxindole exposure repressed growth, with the 50 μM condition showing the weakest biomass accumulation (2.11 mg), followed by the 25 μM group (3.66 mg). The control treatment (R2A without added metabolites) maintained intermediate growth levels. These results confirm that LPA acts as a growth-promoting cue for *Talaromyces*, while oxindole constrains its expansion, aligning with observed metabolomic associations in the native ginger rhizosphere.

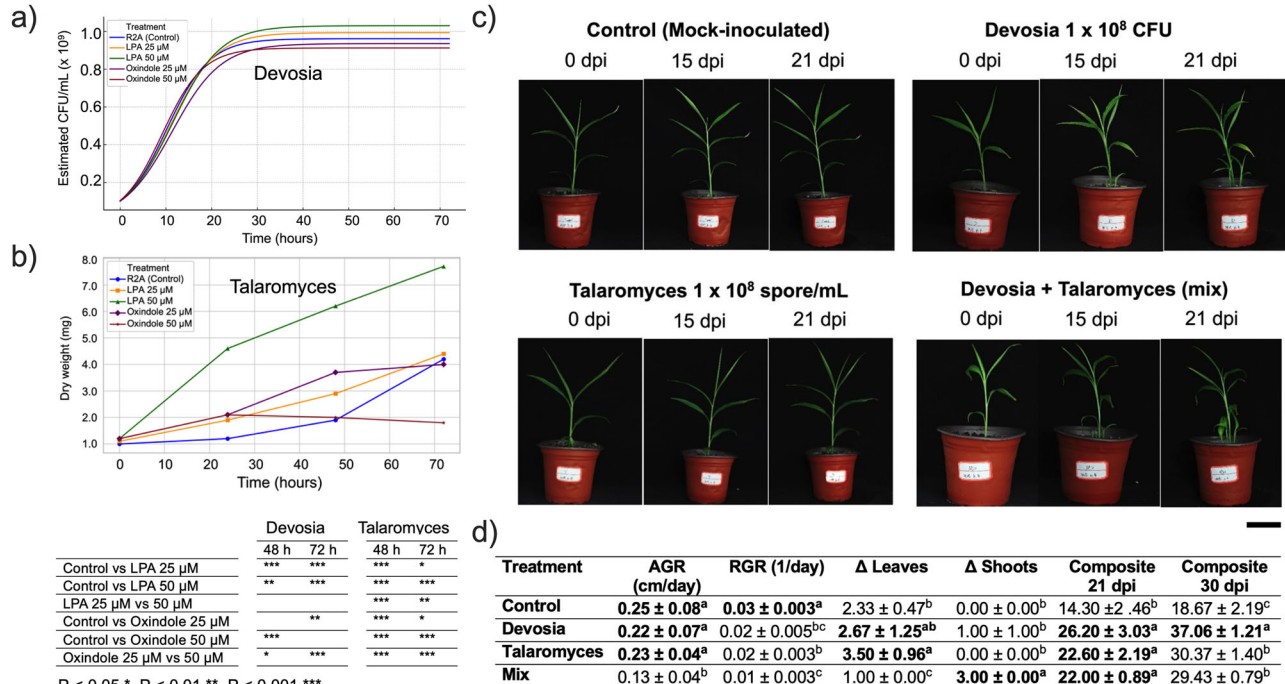

**Fig. 7 | Growth dynamics of keystone microbes in response to metabolites and their effects on ginger seedling development. a** Growth curves of *Devosia riboflavina* ($OD_{600}$ converted to CFU/mL) over 72 h under control conditions and in the presence of 1-oleoyl-LPA (25 μM, 50 μM) or Oxindole (25 μM, 50 μM). **b** Biomass accumulation of *Talaromyces pseudofuniculosus* (dry weight, mg) under the same metabolite treatments over 72 h. Significant differences were observed at 48 and 72 h. **c** Representative images of ginger seedlings at 0, 15, and 21 days post-inoculation (dpi) with four treatments: mock-inoculated control, *Devosia riboflavina* ($1 \times 10^8$ CFU/mL), *Talaromyces pseudofuniculosus* ($1 \times 10^8$ spores/mL), and a microbial consortium (Devosia + Talaromyces). Scale bars, 5 cm. **d** Morphological parameters measured at 21 dpi, including AGR (absolute growth rate), RGR (relative growth rate), ΔLeaves (change in leaf number from 0 to 21 dpi), ΔShoots (net shoot formation), and the composite growth index integrating shoot number, height, and leaf development at 21 dpi. Composite values at 30 dpi were estimated by linear regression. Data represent mean ± s.d. ($n = 6$). Asterisks and different letters indicate statistically significant differences ($p < 0.05$, one-way ANOVA with Tukey's post hoc test).

## Keystone microbial inoculation shapes ginger seedling development

Microbial inoculation significantly influenced several plant growth traits, including absolute growth rate (AGR), relative growth rate (RGR), leaf and shoot development, and composite biomass accumulation at 21 and 30 days post-inoculation (dpi) (Fig. 7c and d). Compared to the control (AGR: 0.25 ± 0.08 cm/day), plants inoculated with *Devosia* (0.22 ± 0.07 cm/day) and *Talaromyces* (0.23 ± 0.04 cm/day) exhibited comparable absolute growth rates ($p > 0.05$). However, co-inoculation with both microbes (Mix) resulted in a significant reduction in AGR (0.13 ± 0.04 cm/day; $p < 0.05$). A similar trend was observed for relative growth rate, where the mix treatment showed the lowest RGR (0.01 ± 0.003 1/day), significantly lower than all other treatments.

*Talaromyces* significantly increased leaf number (Δ Leaves: 3.50 ± 0.96), outperforming both the control (2.33 ± 0.47) and mix treatments (1.00 ± 0.00). Interestingly, the mixed treatment induced the highest shoot proliferation (Δ Shoots: 3.00 ± 0.00), suggesting a potential shift in resource allocation or developmental pattern. At 21 dpi, all inoculated treatments significantly outperformed the control in composite biomass, with *Devosia* leading (26.20 ± 3.03), followed by *Talaromyces* (22.60 ± 2.19) and Mix (22.00 ± 0.89). By 30 dpi, *Devosia*-inoculated plants sustained the highest biomass (37.06 ± 1.21), while *Talaromyces* (30.37 ± 1.40) and Mix (29.43 ± 0.79) showed intermediate performance. The control remained significantly lower (18.67 ± 2.19).

These results suggest that individual microbial strains (*Devosia* and *Talaromyces*) can promote specific aspects of plant growth, with *Talaromyces* enhancing foliar development and *Devosia* contributing to sustained biomass accumulation. In contrast, co-inoculation did not synergise benefits and instead reduced growth rate, highlighting the importance of compatibility and functional precision in microbial consortia design.

## Discussion

The plant microbiome is increasingly acknowledged as a fundamental crop productivity and resilience determinant[28]. However, the ecological and biochemical principles driving its assembly remain insufficiently understood. This study integrates multi-omics approaches to elucidate how host-derived metabolites and microbial interactions co-regulate nutrient cycling and growth in *Zingiber officinale*, offering a roadmap for microbiome-informed crop design.

### Stochastic and deterministic forces in microbiome assembly

Our data reveal distinct assembly dynamics for bacterial and fungal communities across ginger varieties. Bacterial microbiomes fit the neutral model well, and normalised stochasticity ratios were elevated[21,29,30], indicating that community structure is mainly governed by stochastic events such as dispersal and ecological drift[3,5,31]. In contrast, deterministic forces primarily shaped fungal communities, suggesting stronger ecological filtering, possibly mediated by host immune signalling, root exudation profiles, or microhabitat specificity.

The stronger deterministic signature observed in the root and rhizome compartments of the high-yield variety suggests genotype-driven microbial selection. These compartments likely act as ecological filters, enriching for functional taxa involved in nitrogen fixation and pathogen suppression. Such host-mediated filtering may represent an adaptive strategy to promote beneficial microbiome configurations under productivity-enhancing conditions.

### Metabolite-mediated microbial recruitment

Host metabolic fingerprints emerged as significant predictors of microbial composition. Metabolites such as 1-oleoyl lysophosphatidic acid (LPA) and Niazimin A were positively correlated with keystone taxa, including

*Devosia*, *Bradyrhizobium*, and *Sphingomonas*—microbes involved in nitrogen cycling and disease suppression[32,33]. These findings support a model wherein specific plant metabolites act as chemical selectors, orchestrating microbiome assembly by attracting beneficial microbes.

Enriching *Niazimin A, LPA*, and **OoOxindole** in different yield varieties suggests that these metabolites function as *chemical filters*. Niazimin A—a thiocarbamate glycoside from the glucosinolate–myrosinase pathway—has known antimicrobial properties[34]. LPA modulates bacterial membrane fluidity and biofilm formation[35], while Oxindole, a redox-active indole derivative, may influence microbial dynamics via oxidative stress or electron transfer modulation[36].

These compounds likely act as deterministic cues in the rhizosphere, guiding the recruitment of microbial consortia toward beneficial configurations. Notably, potassium and phosphorus emerged as central nutritional drivers, with *Agromyces* as a key mediator. Secondary metabolites, including isoflavonoids and mannose, further shaped microbiome structure through signalling and immune modulation pathways[37–43]. Metabolite–taxon associations also spanned kingdoms, as seen with Oxindole and (S)-5-amino-3-oxohexanoate, suggesting cross-kingdom metabolite signalling[44].

Notably, the same metabolites—LPA and Niazimin A—were also identified in our previous study of healthy versus diseased ginger plants[10,11], associated with beneficial microbes such as *Bacillus* and *Mortierella*. Their repeated detection across health-related and yield-associated contexts reinforces their role as key biochemical signals in microbiome assembly. This consistency suggests that specific host metabolites may act as general selectors of plant-beneficial microbes, regardless of the specific environmental or physiological condition.

## Network structure and keystone taxa
Although microbial complexity does not inherently equate to improved function, intricate co-occurrence networks were associated with the high-yield variety. These networks featured more hub taxa and were enriched in nitrogen-related functions[45–48], suggesting enhanced metabolic redundancy and resilience. Despite low relative abundance, keystone taxa such as *Talaromyces* and *Nitrospira* played central network roles, exemplifying the "rare but vital" paradigm[49–52]. Their presence alongside functionally redundant genera like *Devosia, Sphingomonas*, and *Mortierella*[53–55] may promote ecosystem stability and nutrient turnover.

## Experimental validation of metabolite effects
In vitro metabolite addition assays confirmed the ecological relevance of the selected compounds by demonstrating distinct and consistent effects on the growth dynamics of *Devosia* and *Talaromyces*. LPA acted as a potent microbial growth enhancer across taxa: for *Devosia*, both concentrations of LPA (25 μM and 50 μM) accelerated exponential growth and increased the maximum estimated CFU/mL, while for *Talaromyces*, LPA significantly boosted dry biomass in a dose-dependent manner, with 50 μM achieving the highest yield. Conversely, oxindole elicited inhibitory effects. In *Devosia*, oxindole led to attenuated growth trajectories, with 50 μM causing the most pronounced reduction in carrying capacity. In *Talaromyces*, oxindole severely restricted biomass accumulation, particularly at 50 μM, which resulted in minimal recovery after a sharp 24 h growth dip, suggesting persistent metabolic stress or inhibition of morphogenetic progression.

The sharp decline in *Talaromyces* biomass at 24 h across all treatments (particularly under control and oxindole conditions) may indicate a temporary stress adaptation phase during the transition to liquid culture, possibly involving oxidative or pH changes. The quicker recovery observed with LPA hints at a protective or priming role in fungal metabolism and morphogenesis, aligning with reported involvement of LPA in lipid signalling and stress mitigation pathways[56–59].

These findings offer mechanistic support for the ecological roles inferred from field correlations: LPA may facilitate beneficial taxa by enhancing proliferation and resilience, while oxindole may selectively suppress sensitive or beneficial strains, reinforcing its potential role in niche competition and exclusion[60]. The observed patterns underscore the utility of targeted metabolite assays in validating keystone taxa interactions and metabolic network predictions within plant-associated microbiomes.

## Functional outcomes of inoculation
Contrary to expectations of synergism, individual inoculations with *Devosia* or *Talaromyces* outperformed the combined treatment in promoting growth. *Devosia* enhanced overall biomass, likely through nitrogen fixation or auxin-like effects[61,62], while *Talaromyces* supported leaf expansion. The mix treatment mainly induced shoot formation—possibly via cytokinin-like signalling—but overall biomass and RGR were reduced, perhaps due to competition or antagonism[63–65]. These results challenge the assumption that diversity guarantees functionality. Instead, they support targeted consortia design based on functional compatibility and ecological precision[66].

## Toward rational microbiome engineering
The insights presented here have broad translational potential. Keystone taxa like *Devosia* and *Talaromyces* could be integrated into microbiome formulations for cereals and legumes under low-input regimes[67–69]. Their demonstrated effects on nitrogen and phosphorus cycling offer agronomic value across diverse cropping systems[70]. Small molecules like LPA and Oxindole may also serve as chemical levers to guide microbial recruitment in synthetic communities or breeding programs[71,72]. Future strategies could combine metabolomic trait selection with soil diagnostics and microbial biomarkers for site-specific, genotype-informed crop design.

## Limitations and future directions
Although the two ginger varieties were cultivated in distinct soils and climates, we minimised environmental variation by applying standardised cultivation practices, sampling plants at comparable developmental stages, and maintaining uniform management protocols. These results align with evidence from *Arabidopsis* and maize showing heritable microbial traits and genotype–environment interactions[73–75]. Despite lacking genomic data for the host, the consistent enrichment of functional taxa and variety-specific metabolites suggests a strong host effect. Nevertheless, we cannot entirely exclude the possibility that site-specific factors such as soil type and climate may have contributed to the observed microbial and metabolite differences.

A significant limitation was the taxonomic resolution of fungal ASVs; over 65% remained unclassified—an expected outcome for under-characterised systems like ginger[76–78]. However, consistent beta diversity and network patterns support the robustness of our ecological conclusions. Better resolution through long-read sequencing or metagenomics will improve fungal functional profiling. Combining plant genotype, metabolite cues, and microbial ecology into a unified framework will be crucial for advancing microbiome engineering in precision agriculture.

## Data availability
The datasets generated and analysed in this study are publicly available and comply with community standards for data deposition. Raw sequence data (16S rRNA for bacteria and ITS for fungi) generated via Illumina sequencing have been deposited in the NCBI Sequence Read Archive (SRA) under BioProject accessions PRJNA1035274 (bacteria) and PRJNA1035275 (fungi). Metabolomic datasets are available through the BIG Submission platform (BIG SUB) at the National Genomics Data Centre (NGDC) under Bioproject PRJCA023529 (OMIX ID: OMIX005846; accessible at [https://ngdc.cncb.ac.cn/omix/release/OMIX005846]. The numerical source data underlying all graphs and charts in the leading figures are provided as Supplementary Data files. Specifically, Supplementary Data 1 contains the source data for Fig. 1a–h, Supplementary Data 2 for Fig. 2a–h, Supplementary Data 3 for Fig. 3a–e, Supplementary Data 4 for Fig. 4a–i, Supplementary Data 5 for Fig. 5a–d, Supplementary Data 6 for Fig. 5a–d, and Supplementary Data 7 for Fig. 7a–d. Each spreadsheet includes tabs named according to the figure panels (e.g., Fig 1a, b).

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

## Acknowledgements

This work was supported by the Shandong Province Double Hundred Talent Plan [grant numbers WSG20200001]; the National Natural Science Foundation of China [grant number 32201546]; and the Shandong Province Natural Science Foundation [grant number ZR2022QC138].

## Author contributions

R.S.B. contributed to conceptualisation, methodology, supervision, writing, review, and Editing. N.P.G. contributed to the investigation and writing of the original draft. W.W. contributed to the investigation, funding acquisition, and writing of the original draft. W.H. contributed to the administration and methodology. Y.Z., X.Z., and J.L. contributed to formal analysis. XW contributed to funding acquisition and investigation. B.C. and Y.N. contributed to formal analysis and data curation.

## Competing interests

The authors declare no competing interests.
