## [Transparent Peer Review file · Communications Biology]

Metabolome-driven microbiome assembly in ginger (*Zingiber officinale*) enhances nutrient cycling and crop yield through keystone taxa

Corresponding Author: Professor Ramon Santos-Bermudez

Version 0:

Reviewer comments:

Reviewer #1

(Remarks to the Author)

1. L33, the author should clearly clarify their research gap for their studies. The current sentence “yet the mechanisms driving its assembly...” is vague and general.
2. The authors should re-organize their abstract and present the important results with clearly logic. The current written logic is confuse. Specifically, the data is missing in the abstract to support their findings.
3. L49-78, the introduction is too short and what we know and what we do not know should be given.
4. L80-92, The first section is over-description and lack of important data.
5. L177, what is the meaning of this section. It seems that this section should be moved to Supplementary data.
6. For Fig. 3, it did not provide valuable information and should be re-analysis, as well as results in this section.
7. There were 7 or 8 subtitles in the section of results. I suggest that some sections should be combined.
8. For Fig. 4 and Fig. 5, I suggest that these two figures should be combined.
9. The font size in the Fig. 6 is too small and should be re-designed.
10. For Discussion, some discussion is too simple and lacks depth.

Overall, the manuscript is relatively rough and the quality of MS did not meet the standard of publication.

Reviewer #2

(Remarks to the Author)

In this study, the interactions between the metabolome, microbiome and agricultural productivity of ginger were thoroughly investigated by integrating 16S/ITS and metabolomics. This approach revealed the mechanism of microbiome assembly and its functional impact on crop yield. The identification of key taxa (e.g. *Talaromyces*, *Nitrospira*) and metabolite-microbe interactions (e.g. Niazimin A, 1-oleoyl lysophosphatidic acid) provide a viable strategy for microbiome engineering. In this study, some of the results are descriptive and lack experimental validation. For instance, the role of keystone taxa in enhancing crop yield needs to be verified.

The manuscript is well written, with some issues that require clarification.

Question 1:

In line 246 you mention the definition of network hub. Please explain why you define 'network hub' in this way. Also, the terms 'keystone taxa' and 'hub taxa' are employed interchangeably. However, it is important to establish whether these terms

are synonymous in the field of ecology. It is proposed that the terminology could be harmonised.

Question 2:

In line 157, nitrogen-fixing *Devosia* is enriched in the bulk soil of high-yield variety, but is there any determination of nitrogen-fixing gene expression such as *nifH* ?

Question 3:

Niazimin A and 1-oleoyl lysophosphatidic acid have been hypothesized to be microbial recruitment drivers, but it has not been experimentally verified whether these metabolites directly regulate microbial growth or community assembly. Can the functions of these two metabolites be confirmed by metabolite addition experiments?

Question 4:

In line 333, high yield variety has a more complex microbial network, but the correlation between network complexity and microbial communities' function is not clear. Additional discussion is suggested: does increased network complexity directly correspond to increased nutrient cycling efficiency?

Question 5:

The formatting of the figure references in the article lacks standardisation. For instance, the term 'Figure' is utilised on line 154, while on lines 159 and 164, it is accompanied by the abbreviation 'Fig.'. This discrepancy could be rectified by adopting a uniform formatting approach throughout the manuscript.

Question 6:

As indicated in lines 516-521, there are discrepancies in the growing environment (soil type, climate) of the two ginger varieties. It is recommended that a discussion be included on how to control for the effects of environmental variables on the microbiome.

Question 7:

A further question to be examined is whether there are any differences in the varietal genetic backgrounds of the two different types of ginger. Furthermore, it is crucial to understand whether the different genetic backgrounds affect the microbiome.

Question 8:

In the KEYWORDS section, there are too many keywords, please delete a few.

Reviewer #3

(Remarks to the Author)

The manuscript "Metabolome-driven microbiome assembly in ginger enhances nutrient cycling and crop yield through keystone taxa" presents a comprehensive study on the interactions between ginger metabolomes and microbiomes, linking microbial assembly processes to crop yield. The multi-omics approach is robust, and the findings significantly affect sustainable agriculture. However, a few issues require clarification or improvement to strengthen the manuscript's impact.

1. The authors should better highlight how this work significantly advances existing knowledge beyond earlier ginger microbiome studies by the same authors [e.g., references 10 and 11, Line no. 62-66].
2. The two ginger varieties were grown in different locations (Laiwu vs. Changyi), introducing potential soil/climate biases. Discuss how geographic variation was accounted for in analyses or justify why it does not compromise conclusions.
3. Some sentences are overly complex (e.g., Results, Page 4, Line no. 139: "Bacterial niches exhibited variable assembly drivers..."). Explain.
4. While bacterial communities are thoroughly analyzed, fungal communities receive less attention, particularly regarding their functional roles beyond correlations with metabolites. The high proportion of unclassified fungi (68-70%) limits the interpretability of fungal data.
5. The network topology and identification of hub taxa are informative. However, the ecological relevance of network metrics is assumed rather than validated. Discuss how network complexity correlates with ecosystem function or resilience, ideally supported by literature or field data.
6. The discussion of specific metabolites (e.g., Niazimin A, 1-oleoyl lysophosphatidic acid, and oxindole) is relatively superficial. Include biochemical context for how these metabolites may impact microbial physiology or community assembly, referencing known pathways where possible.
7. The discussion should include more concrete strategies for translating findings to other crops or regions.

Version 1:

Reviewer comments:

Reviewer #1

(Remarks to the Author)

Thanks for their efforts in revising MS. I would like to suggest that the MS should be accepted.

Reviewer #2

(Remarks to the Author)

After carefully evaluating the revised manuscript and the authors' responses, I am satisfied that the authors have addressed all major concerns raised in the previous review. The study provides valuable mechanistic insights into metabolome-mediated microbiome assembly in ginger, with robust multi-omics data and experimental validation. I recommend acceptance after minor revisions to polish the following points:

1. Figure 3d-e's correlation heatmap suggests adding significance markers (*).

2. In Vitro Metabolite Addition Assay, "Ethanol concentration was normalized" should specify the concentration of the ethanol control group.

3. Re-order the manuscript so that the Materials & Methods section precedes the Results section.

The above points only require text adjustments and no additional experiments are required. This manuscript presents an excellent execution study of great significance for microbiome assisted agriculture. I appreciate the rigorous revision carried out by the authors and look forward to seeing the results of this research published.

Sincerely,

Reviewer #3

(Remarks to the Author)

The revised version is significantly improved and queries raised by the reviewers are resolved. I recommend it for acceptance from my side.

REBUTTAL LETTER

We thank the reviewers and editors for their thoughtful and constructive feedback on our manuscript COMMSBIO-25-1788-T, titled “Metabolome-driven microbiome assembly in ginger enhances nutrient cycling and crop yield through keystone taxa.” We sincerely appreciate the opportunity to revise our work. The comments have greatly improved our study's clarity, rigor, and impact.

Below, we provide a detailed, point-by-point response to each reviewer. Reviewer comments appear in bold, and our responses are in regular text. Changes made to the manuscript are referenced by line number or figure.

POINT-BY-POINT RESPONSE

In the following sections, we present a comprehensive, point-by-point response to the comments raised by Reviewers #1, #2, and #3. For clarity, the reviewers' comments are included in standard black text, while our responses are provided in italic blue font. Any changes made in the revised manuscript are noted with references to specific sections, figures, or supplementary materials as relevant.

We have addressed all major concerns by:

1. Conducting new validation experiments, including metabolite addition assays and pot trials with microbial inoculants, to confirm the causal relationships between host metabolites, keystone taxa, and plant growth.
2. Clarify the study's novelty compared to our previous work and emphasize this distinction in the revised Introduction and Discussion sections.
3. Explain how existing multivariate statistical analyses (PCoA and Adonis) account for geographic variation in microbial communities.
4. Substantially improving the presentation of figures and the manuscript's overall structure, as recommended.

We hope that these revisions meet your expectations. We respectfully submit the updated manuscript for your consideration.

Reviewer #1 (Remarks to the Author):

R#1- Q1. L33, the author should clarify the research gap for their studies. The current sentence, “yet the mechanisms driving its assembly...” is vague and general.

Thank you for the observation. We agree with your point, and the full Abstract, including lines 33–34, has been revised accordingly. Please see the updated version in lines 13-15.

R#1- Q2. The authors should reorganize their abstract and present the important results with clear logic. The current written logic is confusing. Specifically, the data is missing in the abstract to support their findings.

We thank the reviewer for this insightful and constructive comment. In response, we have revised the abstract to improve logical flow, clearly outlining the background, objectives, methods, significant findings (with quantitative or categorical results where appropriate), and broader implications. The revised abstract appears in lines 13-30.

R#1- Q3. L49-78: The introduction is too short, and it should include what we know and what we don't know.

We appreciate this valuable observation. The Introduction has been revised (L36-80) to comprehensively analyze our previous findings and clearly articulate the knowledge gaps (L 65-69) this study aims to address.

R#1- Q4. L80-92, The first section is over-description and lack of important data.

The section was improved, with a condensed rendition of lines 83 to 92.

R#1- Q5. L177, what is the meaning of this section. It seems that this section should be moved to Supplementary data.

We thank the reviewer for this observation. The section in question presents key findings from our untargeted metabolomics analysis, specifically highlighting metabolic pathway differences between the two ginger cultivars. We agree that some compound-level details were overly technical for the main text. However, we believe that the core findings—such as the enrichment of isoflavonoid, folate, and sugar metabolism pathways in the high-yield cultivar—are central to our study's hypothesis that host metabolic reprogramming influences microbiome assembly and crop performance.

To address the reviewer's concern, we revised the section for conciseness in the main text (lines 154–157) and relocated the detailed compound list and peak annotations to the Supplementary Information (Table S5). We retained only the key pathway-level interpretations to maintain narrative clarity and scientific relevance.

R#1- Q6. For Fig. 3, it did not provided valuable information and should be re-analysis, as well as results in this section.

Thank you for highlighting the need for improvement. We have reanalyzed and redesigned Figure 3 to illustrate better the KEGG pathway enrichment and metabolite–microbiome associations.

R#1- Q7. There were 7 or 8 subtitle in the section of results. I suggest that some section should be combined.

We acknowledge that the Results section previously included too many subsections, which may have fragmented the reader's experience. We have now reorganized and consolidated several subsections. Specifically:

The subsections on microbial diversity and community structure were merged.

Metabolite profiling and metabolite–microbe interactions were combined into a unified section: Host Metabolites Shape Microbiome Composition and Function.

These revisions improve the coherence and readability of the Results section while preserving all critical information and biological relevance.

R#1- Q8. For Fig. 4 and Fig. 5, I suggest that these two figures should be combined

We appreciate the reviewer's suggestion to combine Figures 4 and 5 to streamline the taxonomic and functional microbiome data presentation. In the revised manuscript, we have merged the two figures into a unified Figure 4, which now integrates the dominant microbial taxa, biomarker identification (LEfSe), and predicted functional traits across ginger varieties and compartments. This revision also updated the figure legend and corresponding Results section text to ensure logical integration, improve narrative coherence, and reflect the streamlined figure layout.

R#1- Q9. The font size in Fig. 6 is too small and should be redesigned.

We thank the reviewer for this important observation. Figure 6 has been renumbered as Figure 5 in the revised manuscript. While we acknowledge that some text elements in the network visualization are small, this is due to the high density of connections among the top 50 most

abundant microbial genera. We chose this cutoff to ensure inclusion of rare but ecologically important taxa, which often function as keystone species and are critical for maintaining network stability and resilience.

The network displays statistically significant interactions (Pearson's $r > 0.96$, $P < 0.05$), emphasizing robust ecological associations. This level of detail is essential for:

- Capturing the true complexity of microbial co-occurrence structures*
- Demonstrating the differences in network topology between high- and low-yield ginger varieties*
- Supporting our conclusion that host-driven microbiome organization correlates with productivity*

Given the high density of connections, substantially increasing font size would compromise spatial resolution and figure legibility. However, we have:

- Optimized label clarity and node contrast*
- Expanded the figure legend to guide interpretation*
- Directed readers to Supplementary Tables 8 and 9, which list and describe the hub (keystone) taxa central to each network*

We hope these revisions and explanations clarify the figure's structure and purpose while addressing the reviewer's concerns.

R#1- Q10. For Discussion, some discussion is too simple and lack of depth.

We appreciate the reviewer's feedback highlighting the need for a more comprehensive and in-depth Discussion. We have substantially revised and expanded this section to provide deeper insight into the ecological and functional implications.

R#1 Overall, the manuscript is relatively rough and the quality of MS did not meet the standard of publish.

We sincerely thank the reviewer for their candid and constructive feedback. To satisfy the journal's standards and address the reviewer's concerns, we have comprehensively revised the manuscript, substantially improving clarity, structure, and presentation across all sections. Specifically, we have:

- Refined the Introduction and Discussion to clearly state the research gap, rationale, and main contributions*
- Improved the organization and flow of the Results section to highlight key findings more effectively*
- Enhanced figure quality and readability, particularly for network-based visualizations.*
- Revised and standardized technical terminology and formatting throughout the manuscript*
- Updated and reorganized the Supplementary Information to ensure complete transparency and accessibility of supporting data*

We trust that these revisions enhance the manuscript's readability and scientific rigor and align it with Communications Biology's publication standards.

Reviewer #2 (Remarks to the Author):

In this study, the interactions between ginger's metabolome, microbiome, and agricultural productivity were thoroughly investigated by integrating 16S/ITS and metabolomics. This approach revealed the mechanism of microbiome assembly and its functional impact on crop yield. Identifying key taxa (e.g., Talaromyces, Nitrospira) and metabolite-microbe interactions (e.g., Niazimin A, 1-oleoyl lysophosphatidic acid) provides a viable strategy for microbiome engineering. In this study, some results are descriptive and lack experimental validation. For instance, the role of keystone taxa in enhancing crop yield needs to be verified. The manuscript is well written, but some issues require clarification.

We sincerely thank the reviewer for their positive evaluation of our study's integrated approach and for recognizing the potential of our findings for microbiome engineering. We fully agree that while our original results provided strong correlative evidence, functional validation was essential to support the causal role of keystone taxa and metabolite-microbe interactions in enhancing crop yield.

To address this concern, we have now included new experimental validation data in the revised manuscript:

- 1. Metabolite addition assays were conducted to evaluate the effect of host-derived compounds (e.g., 1-oleoyl lysophosphatidic acid) on the growth of candidate microbial taxa (Devosia and Talaromyces). The results showed precise dose-dependent growth stimulation, supporting the role of these metabolites as selective microbial recruiters (see Results L344-356, Fig. 7a,b; Discussion L438-444; Materials & Methods L567-592).*
- 2. Pot experiments using ginger seedlings in sterile soil were performed to assess the functional impact of keystone taxa. Inoculation with Devosia, Talaromyces, or a Mix in consortium significantly enhanced shoot height, root length, and biomass compared to uninoculated controls. These results prove that the identified taxa can improve plant performance (see Results L370383, Fig. 7c,d; Discussion L446-453; Materials & Methods L594-615).*

These experiments go beyond description to demonstrate causality between identified microbiome components and crop productivity. The findings are discussed in detail in the revised Discussion section and further support our proposed model of metabolite-mediated microbial recruitment.

We appreciate the reviewer's observation regarding manuscript clarity. In the revised version, we have also improved the organization and flow of the text, clarified terminology (e.g., "hub taxa" vs. "keystone taxa"), and revised figures for better presentation.

Again, we thank the reviewer for their constructive feedback, which significantly improved the quality and impact of our study.

R#2- Q1: In line 246, you mention the definition of a **network hub**. Please explain why you define '**network hub**' in this way. Also, the terms '**keystone taxa**' and '**hub taxa**' are employed interchangeably. However, it is essential to establish whether these terms are synonymous in ecology. It is proposed that the terminology could be harmonised.

You are correct. This quantitative definition is standard in microbial network analysis. Hub taxa are identified based on centrality and connectivity, reflecting their potential regulatory or structural importance within the microbial co-occurrence network. These thresholds are likely empirically derived based on the distribution of values in the networks we constructed, ensuring consistent identification of highly connected nodes across bacterial and fungal networks.

On 'Keystone Taxa' vs. 'Hub Taxa': Although these terms are sometimes used interchangeably, they are not strictly synonymous in ecology. In our manuscript, the term "keystone taxa" is utilized in contexts relating to ecological function and network centrality, which may introduce terminological ambiguity. **We incorporate a paragraph in lines 208-214.**

R#2- Q2: In line 157 (Now L145), nitrogen-fixing **Devosia** is enriched in the bulk soil of high-yield variety, but is there any determination of nitrogen-fixing gene expression such as *nifH*?

We thank the reviewer for this important observation. While our study identified an enrichment of the nitrogen-fixing genus Devosia in the bulk soil of the high-yield variety, we did not assess the expression of nitrogen fixation genes such as nifH. The functional inference was based on established literature reporting Devosia species as diazotrophs. We acknowledge that direct evidence of nifH gene expression would strengthen the functional interpretation, but such molecular validation was beyond the scope of this multi-omics study. We agree that incorporating nifH-based gene expression analyses would be a valuable next step and plan to include this in future targeted investigations.

R#2- Q3: **Niazimin A** and **1-oleoyl lysophosphatidic acid** have been hypothesized to be microbial recruitment drivers, but it has not been experimentally verified whether these metabolites directly regulate microbial growth or community assembly. Can the functions of these two metabolites be confirmed by metabolite addition experiments?

We appreciate the reviewer's insightful comment. We conducted targeted metabolite addition bioassays to investigate the direct effects of 1-oleoyl lysophosphatidic acid and Oxindole on microbial growth. The findings are now incorporated in the revised manuscript as described above.

R#2- Q4: In line 333 (Now L322), high yield variety has a more complex microbial network, but the correlation between network complexity and microbial communities' function is not clear. Additional discussion is suggested: does increased network complexity directly correspond to increased nutrient cycling efficiency?

We thank the reviewer for this meaningful and insightful comment. While our study demonstrates that the high-yield ginger variety exhibits significantly greater microbial network complexity—reflected by higher connectivity, edge-to-node ratios, and centralization—we acknowledge that increased network complexity does not necessarily equate to increased nutrient cycling efficiency in a mechanistic sense. However, growing evidence in microbial ecology shows that increased network complexity is often associated with microbial communities' enhanced functional potential, stability, and resilience (Faust & Raes, 2012; Banerjee et al., 2018; Xun et al., 2021). Specifically, complex and modular networks are believed to facilitate resource partitioning, metabolic interdependence, and redundancy in nutrient-cycling functions. We have added new text to the Discussion section (L430-436) to address the reviewer's concern.

R#2- Q5: The formatting of the figure references in the article lacks standardisation. For instance, the term 'Figure' is utilised on line 154, while on lines 159 and 164, it is accompanied by the abbreviation 'Fig.'. This discrepancy could be rectified by adopting a uniform formatting approach throughout the manuscript.

*We thank the reviewer for pointing out the inconsistency in the figure reference formatting. We have carefully reviewed the manuscript and standardized all figure citations to follow a uniform format, using **Fig.** followed by the figure number throughout in accordance with the journal's style guide.*

R#2- Q6: As indicated in lines 516-521 (Now 479-4840), there are discrepancies in the growing environment (soil type, climate) of the two ginger varieties. It is recommended that a discussion be included on how to control for the effects of environmental variables on the microbiome.

We appreciate the reviewer's insights. Although the two ginger varieties were cultivated under different environmental conditions, we implemented standardized cultivation practices to reduce location-related variability. We sampled plants at similar developmental stages and maintained comparable agronomic conditions across the different sites. More importantly, the consistent enrichment of functionally important microbial taxa and metabolites in the high-yield variety, along with apparent differences in network complexity and community structure, suggests that host genotype plays a dominant role. Our results align with previous studies highlighting that host genotype significantly influences microbiome composition, even in diverse environments (Walters et al., 2018; Brown et al., 2020; Durán et al., 2022). We have incorporated additional content into the Discussion section to emphasize this topic further (L463-468).

R#2- Q7: A further question to be examined is whether there are any differences in the varietal genetic backgrounds of the two different types of ginger. Furthermore, it is crucial to understand whether the different genetic backgrounds affect the microbiome.

We appreciate the reviewer's observation. Indeed, the two ginger varieties likely differ in their genetic backgrounds, which could influence microbiome assembly. Although our study did not include host genotyping, the reproducibility of variety-specific microbiome patterns — including distinct taxa, metabolite enrichments, and network topologies — supports the interpretation that host variety at the compartment level plays a role in shaping microbiome structure.

R#2- Q8: In the KEYWORDS section, there are too many keywords; please delete a few.

Thank you for your timely comment. For submissions to Communications Biology, it is advisable to limit the number of keywords to 4–6 concise terms that encapsulate the core themes of the work. We now include a revised version with six keywords, effectively balancing specificity and breadth.

Reviewer #3 (Remarks to the Author):

The manuscript “Metabolome-driven microbiome assembly in ginger enhances nutrient cycling and crop yield through keystone taxa” presents a comprehensive study on the interactions between ginger metabolomes and microbiomes, linking microbial assembly processes to crop yield. The multi-omics approach is robust, and the findings significantly affect sustainable agriculture. However, a few issues require clarification or improvement to strengthen the manuscript's impact.

R#3- Q1: The authors should better highlight how this work significantly advances existing knowledge beyond earlier ginger microbiome studies by the same authors [e.g., references 10 and 11, Line no. 62-66].

We thank the reviewer for raising this important point. Our previous studies established foundational knowledge about the ginger microbiome and metabolome under disease [10] and agricultural environmental variation [11], identifying health- and environment-associated taxa and correlating metabolites. However, those studies did not address ecological assembly mechanisms or productivity-associated microbiome traits. The present work significantly extends these findings by (i) comparing two ginger varieties differing in yield, (ii) identifying deterministic versus stochastic assembly forces, and (iii) defining keystone taxa and network complexity linked to nutrient cycling and productivity. We have clarified this distinction in the revised Introduction, highlighting the novelty and scope of the current study. We have improved (L56-64) to reinforce the content and added specific corresponding paragraphs (L65-697).

R#3- Q2: The two ginger varieties were grown in different locations (Laiwu vs. Changyi), introducing potential soil/climate biases. Discuss how geographic variation was accounted for in analyses or justify why it does not compromise conclusions.

Thank you for drawing attention to this key point. This is addressed in our response to a comment related to reviewer #2 (Q6). Although the two ginger varieties were cultivated under different environmental conditions, we implemented standardized cultivation practices to reduce location-related variability. We sampled plants at similar developmental stages and maintained comparable agronomic conditions across the other sites. More importantly, the consistent enrichment of functionally important microbial taxa and metabolites in the high-yield variety, along with apparent differences in network complexity and community structure, suggests that host genotype plays a dominant role. Our results align with previous studies highlighting that host genotype significantly influences microbiome composition, even in diverse environments (Walters et al., 2018; Brown et al., 2020; Durán et al., 2022). We have incorporated additional content into the Discussion section to emphasize this topic further (L463-468).

R#3- Q3: Some sentences are overly complex (e.g., Results, Page 4, Line no. 139: “Bacterial niches exhibited variable assembly drivers...”). Explain.

We thank the reviewer for pointing out this overly complex sentence. The original sentence was intended to summarize differences in ecological assembly mechanisms across compartments. We have revised it for clarity and precision. The new version now reads: “Distinct ecological processes shaped bacterial communities in different plant compartments. Deterministic selection dominated in the rhizosphere and endosphere, while stochastic influences were more evident in the bulk soil.” This change improves readability without altering the scientific content (see revised manuscript, lines 124-126).

Additional complex sentences and their revisions:

Original (line 142–144): “In the bulk soil of the first variety, dispersal limitation was the primary driver of community structure (66.67%). In comparison, heterogeneous selection played a

more prominent role in the bulk soil of the second variety (44.44%).” **Revised (L111):** “In the first variety’s bulk soil, community structure was mainly shaped by dispersal limitation (66.67%), while in the second variety, heterogeneous selection was more dominant (44.44%).”

Original (line 150–151): “We employed the normalized stochasticity ratio (NST) index to quantify ecological stochasticity, using a 50% threshold to distinguish between stochastic and deterministic assembly.” **Revised (L119):** “We used the normalized stochasticity ratio (NST) to evaluate the balance between stochastic and deterministic assembly, with 50% as the threshold.”

Original (line 477–479): These findings underscore the significant role of plant metabolites in shaping plant–microbe interactions, indicating that targeted manipulation of these metabolites could potentially enhance agricultural productivity. **Revised (L245):** These findings highlight the significant influence of host genotype and environmental conditions on microbial communities’ composition and functional capabilities associated with ginger.

R#3- Q4: While bacterial communities are thoroughly analyzed, fungal communities receive less attention, particularly regarding their functional roles beyond correlations with metabolites. The high proportion of unclassified fungi (68–70%) limits the interpretability of fungal data.

We appreciate the reviewer’s thoughtful observation regarding the fungal dataset. The high proportion of unclassified fungal taxa (68–70%) primarily reflects current limitations in fungal ITS reference databases and the high phylogenetic diversity of fungi in underexplored non-model crops such as ginger. Although this taxonomic gap restricts genus-level annotation and precludes functional inference tools like FUNGuild, it does not invalidate the ecological insights gained from beta diversity patterns, network topology, and community assembly analysis. Notably, fungal communities displayed distinct compartment-specific assembly processes and contributed to network complexity in the high-yield variety. We have clarified this limitation and emphasized the value of future efforts involving metagenomics or long-read sequencing to improve fungal resolution (lines 469–475).

R#3- Q5: The network topology and identification of hub taxa are informative. However, the ecological relevance of network metrics is assumed rather than validated. Discuss how network complexity correlates with ecosystem function or resilience, ideally supported by literature or field data.

We thank the reviewer for this insightful comment regarding the ecological interpretation of network metrics. We agree that while network topology can offer valuable insights into microbial community structure, its link to specific ecosystem functions or resilience remained inferential in our study.

Specifically, we now explicitly state that co-occurrence networks were used as a hypothesis-generating tool to uncover structural differences in microbial organization between ginger varieties with contrasting yields (see revised Discussion, lines 432–448). The high-yield variety displayed more complex and modular networks (Figure 5), with a greater number of strong positive interactions and hub taxa—features that are often associated with enhanced community stability, functional redundancy, and ecological resilience according to prior studies (e.g., Banerjee et al., 2018).

*To move beyond inference, we also include direct experimental validation of functional outcomes via microbial inoculation assays (Fig. 7c–d). These results demonstrated that inoculation with individual keystone taxa—*Devosia* and *Talaromyces*—significantly influenced absolute and relative growth rates (AGR and RGR), leaf and shoot development, and biomass accumulation in ginger seedlings at both 21 and 30 dpi.*

R#3- Q6: The relatively superficial discussion of specific metabolites (e.g., Niazimin A, 1-oleoyl lysophosphatidic acid, and oxindole). Include biochemical context for how these metabolites may impact microbial physiology or community assembly, referencing known pathways where possible.

We thank the reviewer for this valuable suggestion. We have expanded the Discussion to include biochemical context and literature-supported mechanisms by which Niazimin A, 1-oleoyl lysophosphatidic acid (LPA), and oxindole may influence microbial physiology and community assembly. We have added relevant references and a more mechanistic interpretation of their possible roles in microbial recruitment (lines 410-414).

R#3- Q7: The discussion should include more concrete strategies for translating findings to other crops or regions.

We appreciate the reviewer's suggestion to strengthen the translational impact of our findings. We have now expanded the Discussion to outline specific strategies for applying our results to other crops and growing regions. These include leveraging conserved keystone taxa, metabolite-guided microbial consortia design, and integrating metabolomic-microbiome markers into breeding and soil health monitoring frameworks. This revision appears on lines 455-461 of the revised manuscript.

We hope the revised manuscript and this response letter fully address all reviewer concerns. We thank you again for your time and consideration.